# osl-dynamics, a toolbox for modeling fast dynamic brain activity

Chetan Gohil[1]*, Rukuang Huang[1], Evan Roberts[1], Mats WJ van Es[1], Andrew J Quinn[1,2], Diego Vidaurre[1,3], Mark W Woolrich[1]

[1]Oxford Centre for Human Brain Activity, Wellcome Centre for Integrative Neuroimaging, Department of Psychiatry, University of Oxford, Oxford, United Kingdom; [2]Centre for Human Brain Health, School of Psychology, University of Birmingham, Birmingham, United Kingdom; [3]Center for Functionally Integrative Neuroscience, Department of Clinical Medicine, Aarhus University, Aarhus, Denmark

## eLife assessment

The authors present a comprehensive set of tools to compactly characterize the time-frequency interactions across a network. The utility of the toolbox is **compelling** and demonstrated through a series of exemplar brain imaging datasets. This **fundamental** work adds to the repertoire of techniques that can be used to study high-dimensional data.

*For correspondence:
chetan.gohil@psych.ox.ac.uk

Competing interest: The authors declare that no competing interests exist.

**Abstract** Neural activity contains rich spatiotemporal structure that corresponds to cognition. This includes oscillatory bursting and dynamic activity that span across networks of brain regions, all of which can occur on timescales of tens of milliseconds. While these processes can be accessed through brain recordings and imaging, modeling them presents methodological challenges due to their fast and transient nature. Furthermore, the exact timing and duration of interesting cognitive events are often a priori unknown. Here, we present the OHBA Software Library Dynamics Toolbox (osl-dynamics), a Python-based package that can identify and describe recurrent dynamics in functional neuroimaging data on timescales as fast as tens of milliseconds. At its core are machine learning generative models that are able to adapt to the data and learn the timing, as well as the spatial and spectral characteristics, of brain activity with few assumptions. osl-dynamics incorporates state-of-the-art approaches that can be, and have been, used to elucidate brain dynamics in a wide range of data types, including magneto/electroencephalography, functional magnetic resonance imaging, invasive local field potential recordings, and electrocorticography. It also provides novel summary measures of brain dynamics that can be used to inform our understanding of cognition, behavior, and disease. We hope osl-dynamics will further our understanding of brain function, through its ability to enhance the modeling of fast dynamic processes.

## Introduction

There is growing evidence for the importance of oscillatory activity in brain function (**Buzsáki, 2006**). Neural oscillations have been linked to cognitive processes, such as information encoding and processing, as well as attention (**Ward, 2003**) and distinct oscillatory activity has been observed in different states of consciousness (**Engel and Fries, 2016**). Furthermore, the synchronization of neural oscillations has been proposed as a mechanism for communication (**Fries, 2015**). Neural oscillations have also been a useful tool for understanding brain dysfunction; for example, changes have been observed in the oscillatory activity of diseased and healthy cohorts (**Başar and Güntekin, 2008**).

An aspect of neural oscillations that remains to be fully understood is its dynamic nature, particularly at fast timescales (*van Ede et al., 2018*). Recently, it has been proposed that neuronal populations exhibit short *bursts* of oscillatory activity on timescales of 100 ms (*Jones, 2016*; *Shin et al., 2017*; *Seedat et al., 2020*) rather than the classical view of ongoing oscillations that are modulated in amplitude. This has important implications for how we should be modeling oscillatory activity changes in cognition and disease (*Stevner et al., 2019*; *Khawaldeh et al., 2022*; *Moraud et al., 2018*). Unfortunately, the methods available to detect bursts in oscillatory data are limited, often requiring arbitrary choices for parameters relating to the frequency content, amplitude, and duration (*Bakkum et al., 2013*). These choices can significantly impact the conclusions reached by such analyses.

Furthermore, oscillatory bursting is not isolated to individual brain regions. It has been shown that bursting can occur across cortical networks (*Seedat et al., 2020*), and there are bursts of coherent activity in networks that last on the order of 50–100 ms in both resting-state (*Vidaurre et al., 2018b*) and in task (*Quinn et al., 2018*). Precise knowledge of these fast network dynamics is a valuable insight that can help us understand cognitive processes; for example, the dynamics of specific functional resting-state networks have been linked to memory replay (a≤50 ms process that occurs in memory consolidation) (*Higgins et al., 2021*). Changes in the dynamics of functional networks have also been shown to be predictive of behavioral traits (*Liégeois et al., 2019*) and disease (*Sitnikova et al., 2018*; *Du et al., 2018*; *Van Schependom et al., 2019*; *Salvan et al., 2021*; *Sharma et al., 2021*; *Ma et al., 2020*). The key barrier that prevents us from fully utilizing a network perspective is that the accurate estimation of dynamic functional networks is challenging. This is in part due to the timing and duration of interesting cognitive events, and the corresponding activity in functional networks, not being known. Consequently, we need to rely on methods that can adapt to the data and automatically identify when networks activate.

Here, we present the OHBA Software Library Dynamics Toolbox (osl-dynamics), a Python package that meets two far-reaching methodological challenges that limit the field of cognitive neuroscience: burst detection and the identification of dynamic functional brain networks. It does so by deploying data-driven generative models that have a proven ability to adapt to the data from a wide range of imaging modalities, and can learn the spatiotemporal characteristics of brain activity, with few assumptions and at fast timescales (*Vidaurre et al., 2016*; *Vidaurre et al., 2018a*; *Gohil et al., 2022*).

In applications for burst detection, osl-dynamics can automatically detect oscillatory bursts without the need to specify the frequencies, amplitude threshold, or duration of the bursts. This allows osl-dynamics to answer questions such as: when do oscillatory bursts occur; what is their oscillatory frequency; and what are their characteristic features (e.g. average lifetime, interval, occurrence, and amplitude)?

In the detection of dynamic functional brain networks, osl-dynamics can automatically detect network dynamics at fast timescales with few assumptions. This allows osl-dynamics to answer questions such as: what large-scale functional networks do individuals or groups exhibit; when do these functional networks activate and what are their characteristic dynamics; what functional networks activate in response to a task; do individuals differ in their functional network activations? On top of this, osl-dynamics can characterize functional networks from a more conventional, static (time-averaged), perspective using the same methodology where appropriate as the dynamic methods.

Here, we will illustrate the use of osl-dynamics using publicly available magnetoencephalography (MEG) datasets. However, we emphasize that the scope of the toolbox extends well beyond MEG, containing approaches that can be used, and have been used, to elucidate network and oscillatory dynamics in a range of applications that include electroencephalography (*Hunyadi et al., 2019*; *Ghimatgar et al., 2019*), functional magnetic resonance imaging (fMRI) (*Vidaurre et al., 2018a*; *Vidaurre et al., 2017*), invasive local field potential recordings (*Khawaldeh et al., 2022*; *Garwood et al., 2021*), and electrocorticography (*Wissel et al., 2013*).

## Results

In this section, we outline examples of uses of osl-dynamics to study source reconstructed MEG data. Section 2.1 presents the results of an oscillatory burst analysis pipeline. Sections 2.2–2.4 present the results of various dynamic network analysis pipelines. For comparison, we also include the results of a static network analysis pipeline in Section 2.5. For a description of the methods and dataset see Section 5.

## 2.1 Burst detection using a single-region TDE-HMM

The pipeline in Figure 10 was applied to do burst detection on a single parcel in the left motor cortex. The source data was calculated using the CTF rest MEG dataset. All subjects were concatenated temporally and used to train the TDE-HMM. The results are shown in *Figure 1*.

We see from the wavelet transform in *Figure 1AI* that there are short bursts of oscillatory activity in this time series. This illustrates how it would be non-trivial, using conventional bandpass filtering and thresholding methods, to identify when exactly a burst occurs and what frequencies are contained within it. Instead of a conventional burst detection method, we use a three state TDE-HMM to identify bursts in a data-driven fashion. We see from the inferred state probability time course (*Figure 1BI*) that there are short-lived states that describe this data. We can see from *Figure 1BII* that each state corresponds to unique oscillatory activity. State 1 is interpreted as a non-oscillatory background state because it does not show any significant peaks in its PSD. States 2 and 3 show oscillatory activity in the $\delta/\theta$ band (1–7 Hz) and $\alpha/\beta$ band (7–30 Hz), respectively. *Figure 1BIII* shows the correlation of each state probability time course with the AEs for different frequency bands (*Figure 1AII*). Based on this, we identify state 2 as a $\delta/\theta$-burst state and state 3 as a $\beta$-burst state. We can see from *Figure 1BIV* that these bursts have a variety of lifetimes ranging

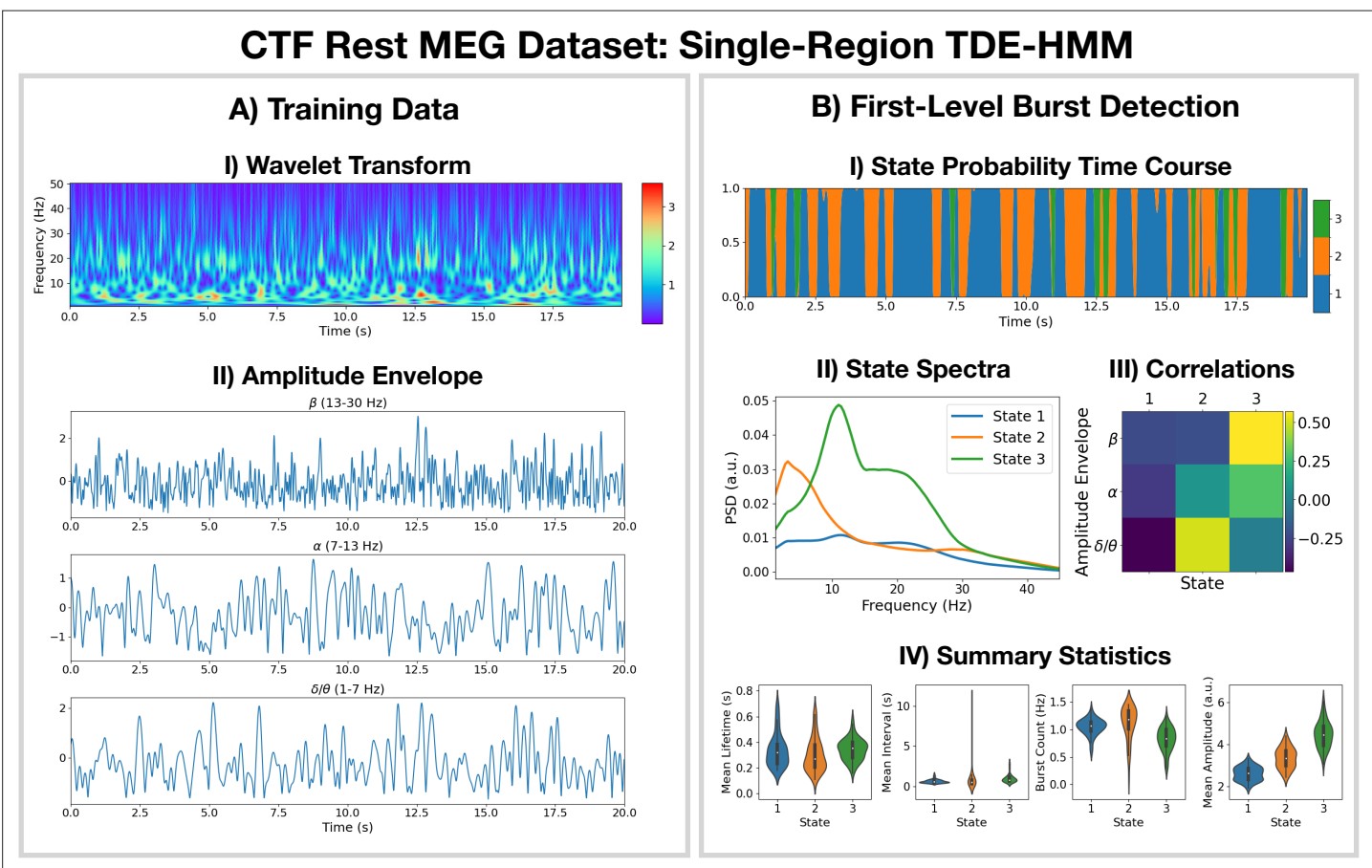

**Figure 1.** Burst detection: single region source reconstructed magnetoencephalography (MEG) data (left motor cortex) shows short-lived bursts of oscillatory activity. (**A.I**) Dynamic spectral properties of the first 20 s of the time series from the first subject. (**A.II**) Amplitude envelope calculated after bandpass filtering the time series over the $\beta$-band (top), $\alpha$-band (middle), and $\delta/\theta$-band (bottom). (**B.I**) The inferred state probability time course for the first 20 s of the first subject. (**B.II**) The power spectral density (PSD) of each state. (**B.III**) Pearson correlation of each state probability time course with the amplitude envelopes for different frequency bands. (**B.IV**) Distribution over subjects for summary statistics characterizing the bursts. Note that no additional bandpass filtering was done to the source data when calculating the mean amplitude. The script used to generate the results in this figure is here: https://github.com/OHBA-analysis/osl-dynamics/blob/main/examples/toolbox_paper/ctf_rest/tde_hmm_bursts.py; (*Gohil, 2024*).

The online version of this article includes the following figure supplement(s) for figure 1:

**Figure supplement 1.** Reproducibility of the CTF rest magnetoencephalography (MEG) dataset single-region TDE-HMM analysis.

from a hundred to several hundred milliseconds. The reproducibility of these results is shown in *Figure 1—figure supplement 1*.

## 2.2 Detecting network dynamics using a multi-region AE-HMM

The AE-HMM pipeline in Figure 11 was applied to source reconstructed data from the Elekta task MEG dataset to identify amplitude-based network dynamics. All subjects and runs were concatenated temporally to train the model. The results are shown in *Figure 2* with an example of a group-level analysis on the HMM state time courses (calculation of a group-averaged network response).

We see the AE-HMM identifies plausible functional networks (*Buckner et al., 2011*) with fast dynamics, typically with lifetimes of around 50 ms (*Figure 2AIII*). We identify a default mode network (state 1); two visual networks (states 2 and 6); a frontotemporal networks (states 3 and 7); and a sensorimotor network (state 4).

The AE-HMM was trained on the continuous source reconstructed data in an *unsupervised* manner, i.e., without any knowledge of the task. Post-HMM training, we can epoch the inferred state time course (Veterbi path) around the task (presentation of a visual stimuli [A picture of a familiar, unfamiliar, or scrambled face]) and average over trials. This gives the probability of each state being activate around a visual event. This is shown in *Figure 2BI*. We observe a significant increase (p<0.05) in the activation of the visual networks (states 2 and 6) between 50–100 ms after the presentation of the visual stimuli as expected. We also observe a significant activation (p<0.05) of the frontotemporal network (state 7) 300–900 ms after the visual stimuli as well as a deactivation of the visual networks (states 2 and 6). The reproducibility of these results is shown in *Figure 2—figure supplement 1*.

## 2.3 Detecting network dynamics using a multi-region TDE-HMM

The TDE-HMM pipeline in Figure 11 was also applied to the Elekta task MEG dataset. All subjects and runs were concatenated temporally and used to train the model. The results are shown in *Figure 3*.

For the high-power networks (states 1–4), we see the same spatial patterns in TDE-HMM power maps (*Figure 3AI*, top) and AE-HMM amplitude maps (*Figure 2AI*). We can see from the state PSDs (*Figure 3AI*, bottom) that the networks identified by the TDE-HMM exhibit distinct spectral (oscillatory) activity. The TDE-HMM networks also have fast dynamics (*Figure 3AIII*) with lifetimes of around 50 ms. In *Figure 3BI*, we can see we are able to reproduce the network response analysis we did using the AE-HMM (*Figure 2BI*). The reproducibility of these results is shown in *Figure 3—figure supplement 1*.

The Elekta MEG dataset was recorded during a visual perception task. For comparison, we perform the same analysis on the CTF rest MEG dataset. All subjects were concatenated temporally and used to train the model. *Figure 4* shows the results of applying a TDE-HMM pipeline to this dataset. We observe similar networks in rest (*Figure 4A*) as in task (*Figure 3A*), which is a known result from fMRI studies (*Smith et al., 2009*). We also include the coherence networks in *Figures 3AI and 4AI*. We observe regions with high power activations have high connectivity (coherence). These networks also have fast dynamics (*Figure 4AIII*) with lifetimes of 50–100 ms.

To illustrate a group-level analysis we could do with a dynamic network perspective, we compared two groups: 27 subjects in a young group (18–34 years old) and 38 subjects in an old group (34–60 years). *Figure 4BI*.I shows summary statistics for each group. We see the fractional occupancy and switching rate of the sensorimotor network (state 4) is increased in the older group (p<0.05). The mean lifetime of the visual network (state 6) is also decreased in the older group (p<0.05). The older group also has a wider distribution of mean intervals for the default mode network (state 1) and suppressed state (8) (p<0.05). The reproducibility of these results is shown in *Figure 4—figure supplement 1*. The age-related differences we observe here are consistent with existing studies (*Coquelet et al., 2020*). We will discuss the young vs old comparison further in Section 2.5.

## 2.4 Dynamic network detection using multi-region TDE-DyNeMo

The TDE-DyNeMo pipeline in Figure 11 was applied to the CTF rest MEG dataset. All subjects were concatenated temporally and used to train the model. The results are shown in *Figure 5*. Note, for DyNeMo we found that learning seven modes (rather than 8) led to more reproducible results. Therefore, we present the 7-mode fit in *Figure 5*.

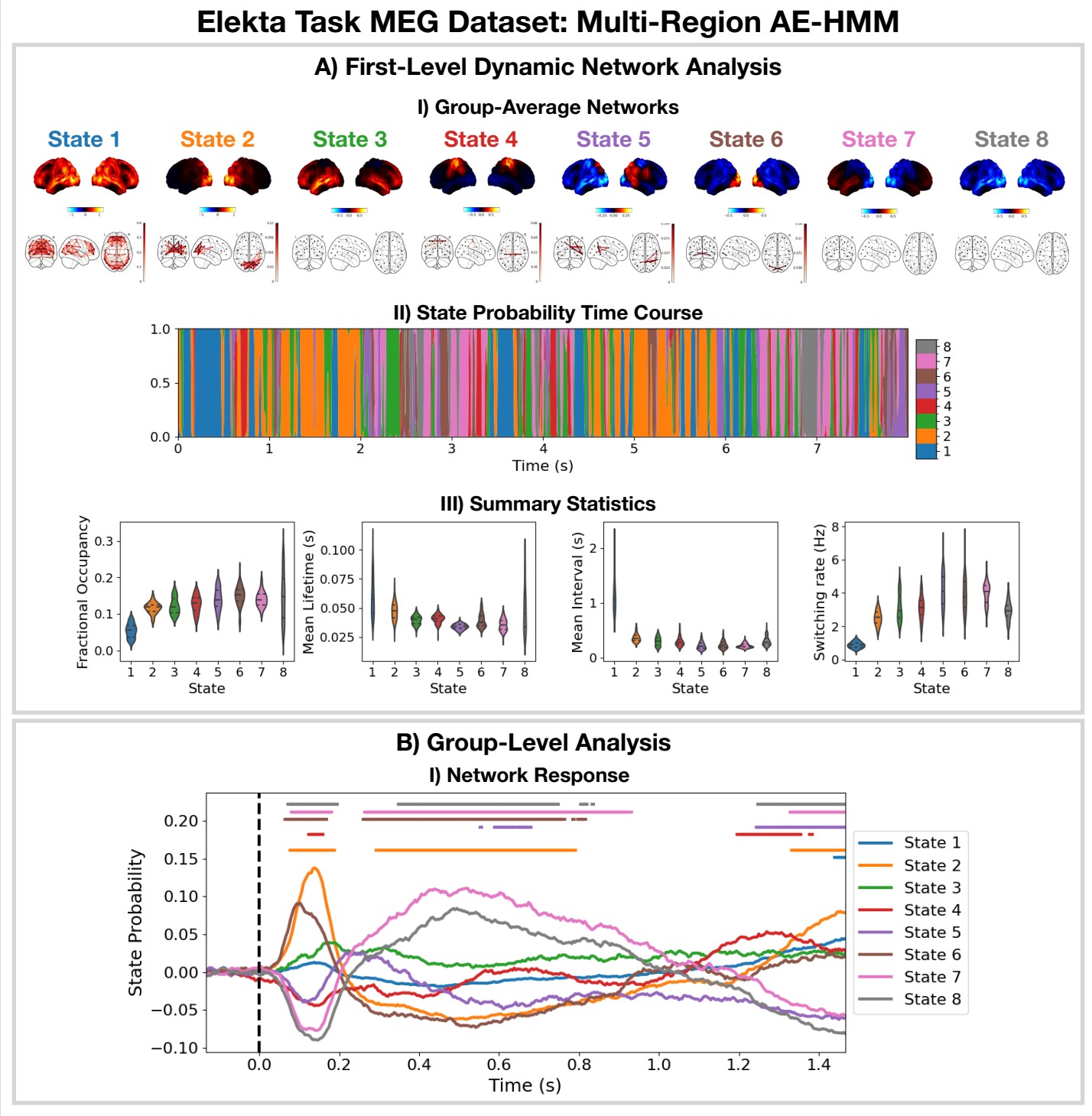

**Figure 2.** Dynamic network detection: a multi-region AE-HMM trained on the Elekta task magnetoencephalography (MEG) dataset reveals functional networks with fast dynamics that are related to the task. (**A.I**) For each state, group-averaged amplitude maps relative to the mean across states (top) and absolute amplitude envelope correlation networks (bottom). (**A.II**) State probability time course for the first 8 s of the first subject. (**A.III**) Distribution over subjects for the summary statistics for each state. (**B.I**) State time courses (Viterbi path) epoched around the presentation of visual stimuli. The horizontal bars indicate time points with p<0.05. The maximum statistic pooling over states and time points was used in permutation testing to control for the family-wise error rate. The script used to generate the results in this figure is here: https://github.com/OHBA-analysis/osl-dynamics/blob/main/examples/toolbox_paper/elekta_task/ae_hmm.py.

The online version of this article includes the following figure supplement(s) for figure 2:

**Figure supplement 1.** Reproducibility of the Elekta task magnetoencephalography (MEG) dataset multi-region AE-HMM analysis.

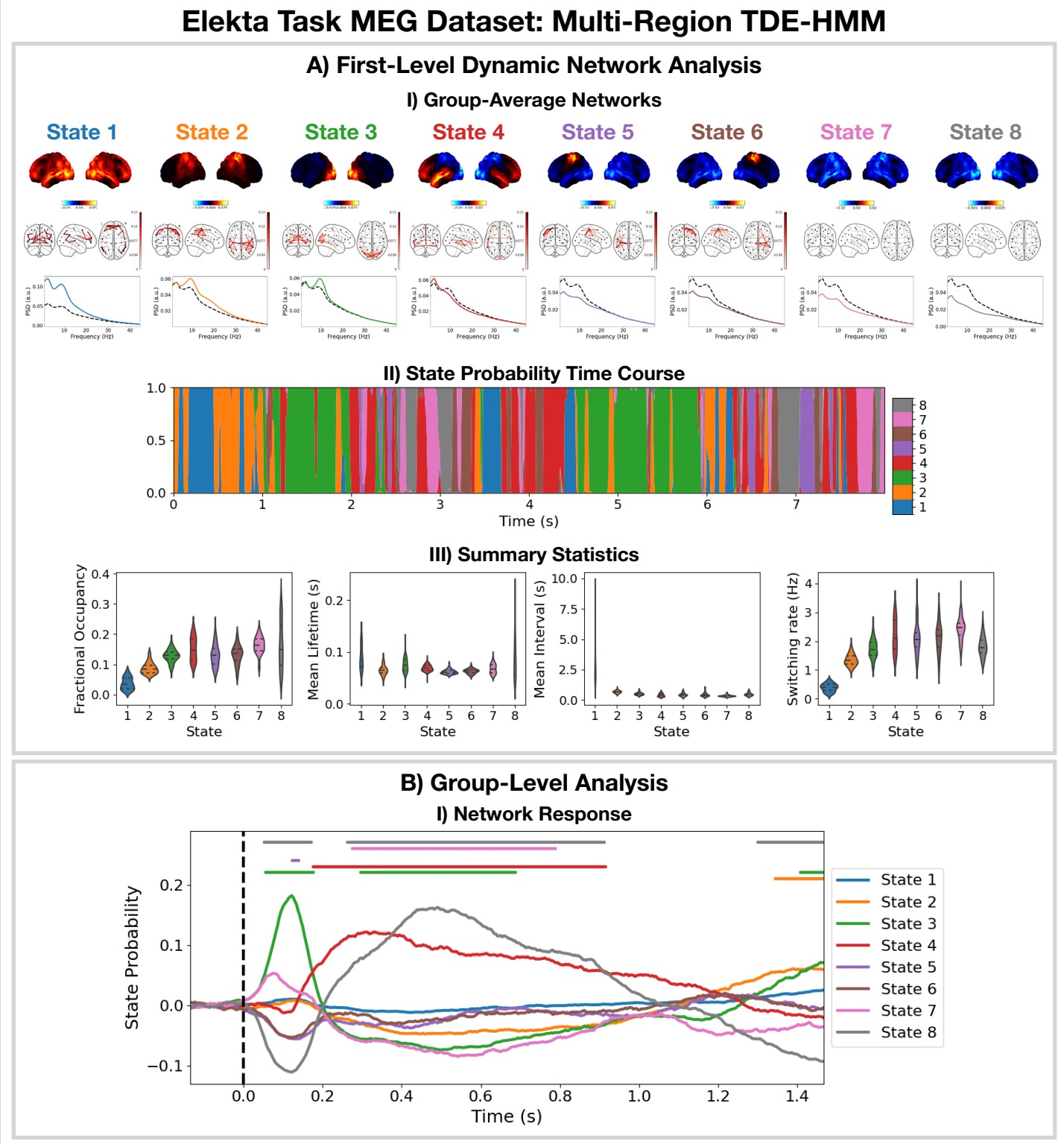

**Figure 3.** Dynamic network detection: a multi-region TDE-HMM trained on the Elekta task magnetoencephalography (MEG) dataset reveals spectrally distinct functional networks with fast dynamics. (**A.I**) For each state, group-averaged power maps relative to the mean across states (top), coherence networks (middle), and power spectral density (PSD) averaged over regions (bottom), both the state-specific (colored solid line), and static PSD (i.e. the average across states, dashed black line) are shown. (**A.II**) State probability time course for the first 8 s of the first subject. (**A.III**) Distribution over subjects for the summary statistics for each state. (**B.I**) State time courses (Viterbi path) epoched around the presentation of visual stimuli. The horizontal bars indicate time points with p<0.05. The maximum statistic pooling over states and time points was used in permutation testing to control

*Figure 3 continued on next page*

*Figure 3 continued*

for the family-wise error rate. The script used to generate the results in this figure is here: https://github.com/OHBA-analysis/osl-dynamics/blob/main/examples/toolbox_paper/elekta_task/tde_hmm.py.

The online version of this article includes the following figure supplement(s) for figure 3:

**Figure supplement 1.** Reproducibility of the Elekta task magnetoencephalography (MEG) dataset multi-region TDE-HMM analysis.

We can see from the power maps and coherence networks (*Figure 5AI*) that DyNeMo identifies much more localized power activations and a cleaner network structure than was seen with the TDE-HMM. We can see from the PSDs (*Figure 5AI*, bottom) that these networks also exhibit distinct spectral characteristics.

From the (renormalized) mode time course (*Figure 5AII*) we see the description provided by DyNeMo is that of overlapping networks that dynamically vary in mixing ratios. This is a complementary perspective to the state description provided by the HMM. Co-activations of each mode can be understood by looking at the Pearson correlation between (renormalized) mode time courses (*Figure 5AIII*). We observe modes with activity in neighboring regions show more co-activation. We summarize the (renormalized) mode time course using statistics (the mean and standard deviation) in *Figure 5AIV*.

To compare DyNeMo to the HMM in a group-level analysis, we repeat the young vs old study using the DyNeMo-specific summary statistics (i.e. the mean and standard deviation of the renormalized mode time courses). *Figure 5BI* shows significant group differences for young (18–34 years old) and old (34–60 years old) participants. We can see an increased mode contribution (mean renormalized mode time course) for the sensorimotor network (mode 4), which reflects the increase in fractional occupancy we saw in the TDE-HMM (*Figure 4BI*). We see DyNeMo is able to reveal a stronger effect size with a $p < 0.01$ compared to the TDE-HMM, which had a $p < 0.05$. DyNeMo also shows a decrease in the variability (standard deviation of the renormalized mode time course) for the left temporal network (mode 5, $p < 0.01$). We will discuss the young vs old comparison further in Section 2.5. The reproducibility of these results is shown in *Figure 5—figure supplement 1*.

## 2.5 Estimating static functional networks

For comparison, we also apply a typical static network analysis pipeline (including static functional connectivity) to the CTF rest MEG dataset. We also consider how the static perspective in a young vs old group-level analysis compares to the dynamic perspective provided by the TDE-HMM in *Figure 4* and TDE-DyNeMo in *Figure 5*, illustrating the benefits of being able to do static and dynamic analyses within the same toolbox.

*Figure 6* shows the group-averaged PSD (A.I), power maps (A.II), coherence networks (A.III) and amplitude envelope correlation (AEC) networks (A.IV) were calculated using all subjects. We observe $\delta$-power is strongest in anterior regions and $\alpha$-power is strongest in posterior regions. We also observe qualitatively similar coherence and AEC networks. In particular, we see strong occipital connectivity in the $\alpha$-band in both the coherence and AEC networks.

*Figure 6B* shows significant ($p < 0.05$) differences in the power maps (B.I) and AEC networks (B.II) for old (34–60 years old) minus young (18–34 years old) groups. We observe a significant reduction in temporal $\delta$-power and an increase in sensorimotor $\beta$-power. We also observe a significant increase in sensorimotor AEC in the $\beta$-band (*Figure 6BII*).

The static (time-averaged) differences we see in young vs old participants can arise in many ways from the underlying dynamics of resting-state networks (*Figures 4AI and 5AI*). For example, an increase in static power could be due to more frequent activations of a particular network. Conversely, the dynamics of the networks may be unaffected and the power within a network could be altered. Studying the dynamic network perspective using the HMM and/or DyNeMo can help provide further insights into how the static differences arise. Looking at the dynamic network perspective provided by the TDE-HMM, we see an increase in the fractional occupancy of state 4 (*Figure 4BI*), which is a network with high $\beta$-power and connectivity (coherence) in the sensorimotor region. This is consistent with the static increase in $\beta$-power and AEC connectivity we observe here; i.e., the increase in static $\beta$-power and connectivity with age can be linked to a larger fraction of time spent in the sensorimotor network. The perspective provided by TDE-DyNeMo shows an increase with age in the contribution

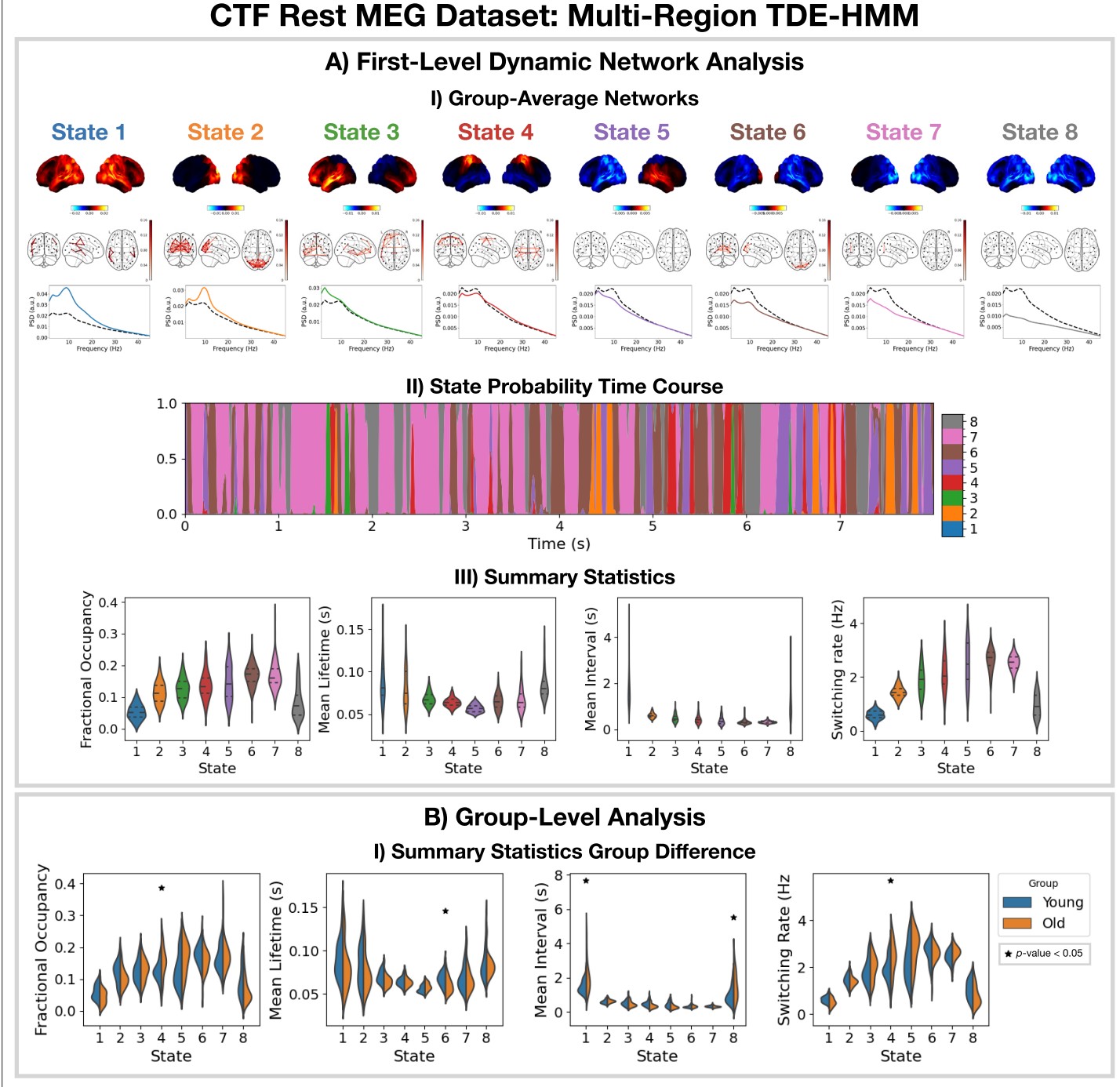

**Figure 4.** Dynamic network detection: a multi-region TDE-HMM trained on the CTF rest magnetoencephalography (MEG) dataset identifies the same functional networks to those found with the Elekta task MEG dataset and reveals differences in the dynamics for young vs old groups. (**A.I**) For each state, group-averaged power maps relative to mean across states (top), absolute coherence networks (middle) and power spectral density (PSD) averaged over regions (bottom), both the state-specific (colored solid line) and static PSD (i.e. the average across states, dashed black line) are shown. (**A.II**) State probability time course for the first 8 s of the first subject and run. (**A.III**) Distribution over subjects for the summary statistics of each state. (**B.I**) Comparison of the summary statistics for a young (18–34 years old) and old (34–60 years old) group. The star indicates a p<0.05. The maximum statistic pooling over states and metrics was used in permutation testing to control for the family-wise error rate. The script used to generate the results in this figure is here: https://github.com/OHBA-analysis/osl-dynamics/blob/main/examples/toolbox_paper/ctf_rest/tde_hmm_networks.py; (*Gohil, 2024*).

The online version of this article includes the following figure supplement(s) for figure 4:

**Figure supplement 1.** Reproducibility of the CTF rest magnetoencephalography (MEG) dataset multi-region TDE-HMM analysis.

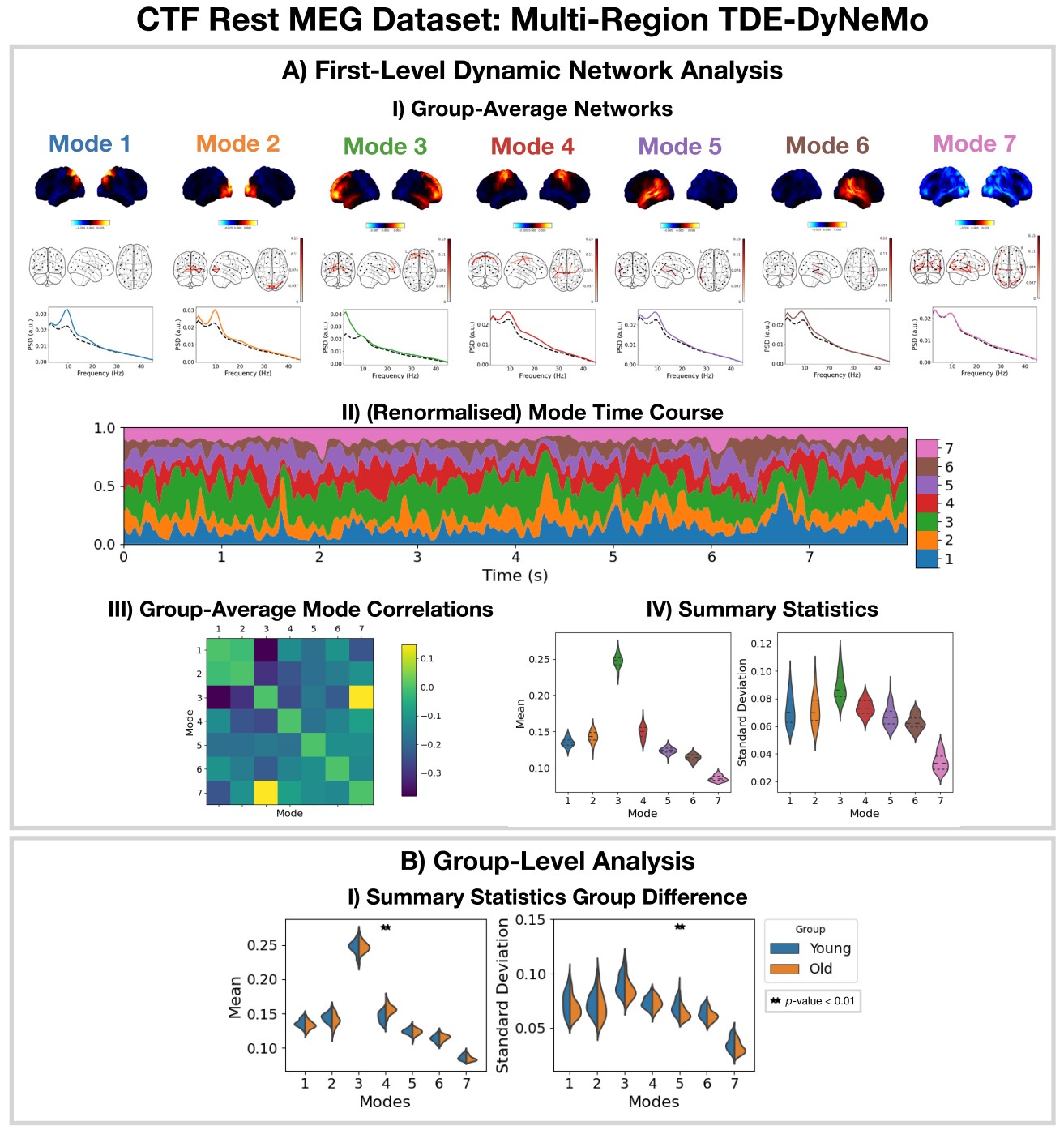

**Figure 5.** Dynamic network detection: a multi-region TDE-DyNeMo trained on the CTF rest magnetoencephalography (MEG) dataset reveals spectrally distinct modes that are more localized than Hidden Markov Model (HMM) states and overlap in time. (**A.I**) For each mode, group-averaged power maps relative to the mean across modes (top), absolute coherence networks (middle), and power spectral density (PSD) averaged over regions (bottom), both the mode-specific (colored solid line) and static PSD (i.e. the average across modes, dashed black line) are shown. (**A.II**) Mode time course (mixing coefficients) renormalized using the trace of the mode covariances. (**A.III**) Pearson correlation between renormalized mode time courses calculated by concatenating the time series from each subject. (**A.IV**) Distribution over subjects for summary statistics (mean and standard deviation) of the renormalized mode time courses. (**B.I**) Comparison of the summary statistics for a young (18–34 years old) and old (34–60 years old) group. The maximum statistic pooling over modes and metrics was used in permutation testing to control for the family-wise error rate. The script used to generate the results in this figure is here: https://github.com/OHBA-analysis/osl-dynamics/blob/main/examples/toolbox_paper/ctf_rest/tde_dynemo_networks.py.

*Figure 5 continued on next page*

*Figure 5 continued*

The online version of this article includes the following figure supplement(s) for figure 5:

**Figure supplement 1.** Reproducibility of the CTF rest magnetoencephalography (MEG) dataset multi-region TDE-DyNeMo analysis.

of mode 4 (*Figure 5BI*), which represents a sensorimotor network. This is a complementary explanation for the increase in static $\beta$-power and connectivity as a larger contribution from the sensorimotor network to the overall brain activity of older participants.

## Discussion

In Section 2.1, we use the TDE-HMM to identify oscillatory bursts in a data-driven manner with much fewer assumptions than conventional burst detection methods based on amplitude thresholding. The advantages of using a data-driven approach like the TDE-HMM are discussed further in *Seedat et al., 2020*; *Masaracchia, 2023*. In short, with a conventional approach, we must pre-specify a frequency of interest and we may miss oscillatory bursts that do not reach an arbitrary threshold. In contrast, the TDE-HMM is less sensitive to the amplitude (it is better able to identify low-amplitude oscillatory bursts) and can identify the frequency of oscillations automatically.

In Sections 2.2 and 2.3, we presented the functional networks identified by HMMs in a variety of settings. These networks were identified automatically at fast (sub-second) timescales from the data (unsupervised) with no input from the user. We found a set of plausible networks that were related to task (*Figure 2*) and demographics (*Figure 4*). These networks were very reproducible: across multiple HMM runs; across different data preparation techniques (AE and TDE); across different experimental paradigms (task and rest); and across different scanners (Elekta and CTF).

Given we observe similar networks with the AE-HMM and TDE-HMM (*Figures 2 and 3*, respectively), one may ask which pipeline is recommended. The TDE-HMM approach is able to model dynamics in oscillatory amplitude and phase synchronization whereas the AE-HMM can only model dynamics in amplitude. This means the TDE-HMM is generally a better model for oscillatory dynamics. An occasion where the AE-HMM may be preferred is if the extra computational load of training on TDE/PCA data prohibits the TDE-HMM.

An important choice that has to be made when training an HMM/DyNeMo is the number of states/modes. For burst detection, we are often interested in identifying the time points when a burst occurs. This can be achieved by fitting a two-state HMM: an 'on' and 'off' state. If we're interested in multiple burst types, we can increase the number of states. In this work, we chose a three-state HMM to stay close to the on/off description while allowing for multiple burst types. For the dynamic network analysis, we want a low number of states/modes to give us a compact representation of the data. A common choice is between 6 and 12. We can use the reproducibility analysis (section 2 in the SI) to show a given number of states/modes is reproducible and use this to find an upper limit for the number of states/modes that can be reliably inferred.

osl-dynamics offers a choice of two generative models for detecting network dynamics: the HMM and DyNeMo. The HMM assumes that there are mutually exclusive network states, whereas DyNeMo assumes the network modes are mixed differently at each time point. While DyNeMo's assumption is arguably more realistic, the HMM's stronger assumption has the benefit of simplifying the decomposition, which can make interpreting the network dynamics more straightforward. In short, the HMM and DyNeMo provide complementary descriptions of network dynamics, with either one being potentially useful depending on the context (*Gohil et al., 2022*). DyNeMo does have the additional advantage of using a richer temporal regularisation through the use of a deep recurrent network. This has been shown to capture longer-range temporal structure than the HMM (*Gohil et al., 2022*), and exploring the cognitive importance of long-range temporal structure is an interesting area of future investigation (*van Es, 2023*). It is possible to quantify which model is better using a downstream task. In this manuscript, the downstream tasks are the evoked network response and young vs old group differences. We argue a better performance in the downstream task indicates a more useful model.

osl-dynamics can also be used to compute static network descriptions, including conventional static functional connectivity. This uses the same methodology as the state (or mode) specific network estimation in the dynamic approaches, making comparisons between dynamic and static perspectives more straightforward. In Section 2.5, we used this feature to relate the static functional network

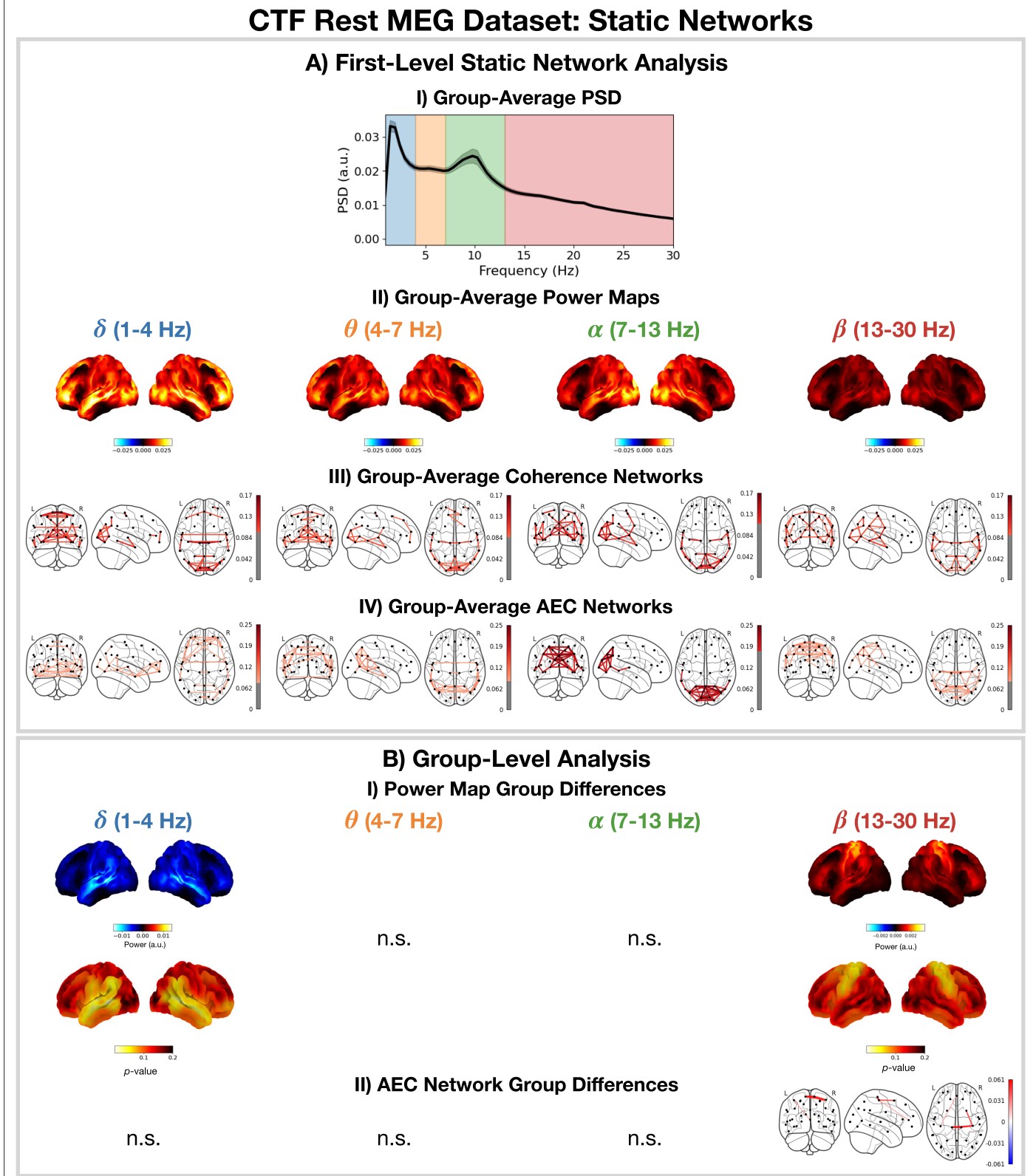

**Figure 6.** Static network detection: osl-dynamics can also be used to perform static network analyses (including functional connectivity). In the CTF rest magnetoencephalography (MEG) dataset, this reveals frequency-specific differences in the static functional networks of young (18–34 years old) and old (34–60 years old) participants. Group-average power spectral density (PSD) (averaged over subjects and parcels, (**A.I**) power maps (**A.II**) coherence networks (**A.III**) and AEC networks (**A.IV**) for the canonical frequency bands ($\delta, \theta, \alpha, \beta$). (**B.I**) Power difference for old minus young (top) and p-values

*Figure 6 continued on next page*

*Figure 6 continued*

(bottom). Only frequency bands with at least one parcel with a p<0.05 are shown, the rest are marked with n.s. (none significant). B.II) Amplitude envelope correlation (AEC) difference for old minus young only showing edges with a p<0.05. The maximum statistic pooling over frequency bands and parcels/edges was used in permutation testing to control for the family-wise error rate. The script used to generate the results in this figure is here: https://github.com/OHBA-analysis/osl-dynamics/blob/main/examples/toolbox_paper/ctf_rest/static_networks.py.

description to a dynamic perspective. We would like to stress that the young vs old study is used as an example of the type of group analyses that can be performed with this toolbox and that a more rigorous study with a larger population dataset is needed to understand the impact of aging on functional networks. The results in Section 2.5 should be taken as just an indication of possible ageing effects that can be investigated in a future study. In this report, we focus on the presentation of the tools needed to make such studies possible.

## Conclusions

We present a new toolbox for studying time series data: osl-dynamics. This is an open-source package written in Python. We believe the availability of this package in Python improves the accessibility of these tools, in particular for non-technical users. Additionally, it avoids the need for a paid license. Using Python also enables us to take advantage of modern deep learning libraries (in particular *TensorFlow, 2023*) which enables us to scale these methods to very large datasets, something that is currently not possible with existing toolboxes.

osl-dynamics can be used, and has been used, in a wide range of applications and on a variety of data modalities: electrophysiological, invasive local field potential, functional magnetic resonance imaging, etc. Here, we illustrated its use in applications of burst detection and dynamic network analysis using MEG data. This package also allows the user to study the static (time-averaged) properties of a time series alongside dynamics within the same toolbox. The methods contained in osl-dynamics provide novel summary measures for dynamics and group-level analysis tools that can be used to inform our understanding of cognition, behavior, and disease.

## Materials and methods
### 5.1 Generative models

In the study of dynamics, we are often interested in the properties of a time series, such as power spectral density (PSD), mean, covariance, etc., at a given time point. A common heuristic approach for calculating this is to use a sliding window. However, this approach only utilizes a short window around the time point of interest and suffers from a tradeoff between the temporal precision of dynamics and an accurate estimation of the properties (via a sufficiently large window). In *Liuzzi et al., 2019*, it was shown that this approach is inadequate for studying fast changes in functional connectivity. In osl-dynamics, we adopt an alternative approach based on *generative models* (*Lamb, 2021*). These are models that learn the probability distribution of the training data. In this report, we will focus on two generative models for time series data: the Hidden Markov Model (HMM) (*Rezek and Roberts, 2005*) and Dynamic Network Modes (DyNeMo) (*Gohil et al., 2022*). Both of these models (discussed further below) incorporate an underlying dynamic latent variable in the generative process. The objective during training is to learn the most likely latent variables to have generated the observed time series (we minimize the *variational free energy* [This is equivalent to maximizing the negative variational free energy, which is also known as the evidence lower bound (ELBO) *Bishop and Nasrabadi, 2006*] *Bishop and Nasrabadi, 2006*). In doing this, the model can identify non-contiguous segments of the time series that share the same latent variable. Pooling this information leads to more robust estimates of the local properties of the data.

The generative model for the HMM (shown in *Figure 7A*) is

$$p(\boldsymbol{x}_{1:T}, \theta_{1:T}) = p(\boldsymbol{x}_1|\theta_1)p(\theta_1)\prod_{t=2}^{T} p(\boldsymbol{x}_t|\theta_t)p(\theta_t|\theta_{t-1}), \tag{1}$$

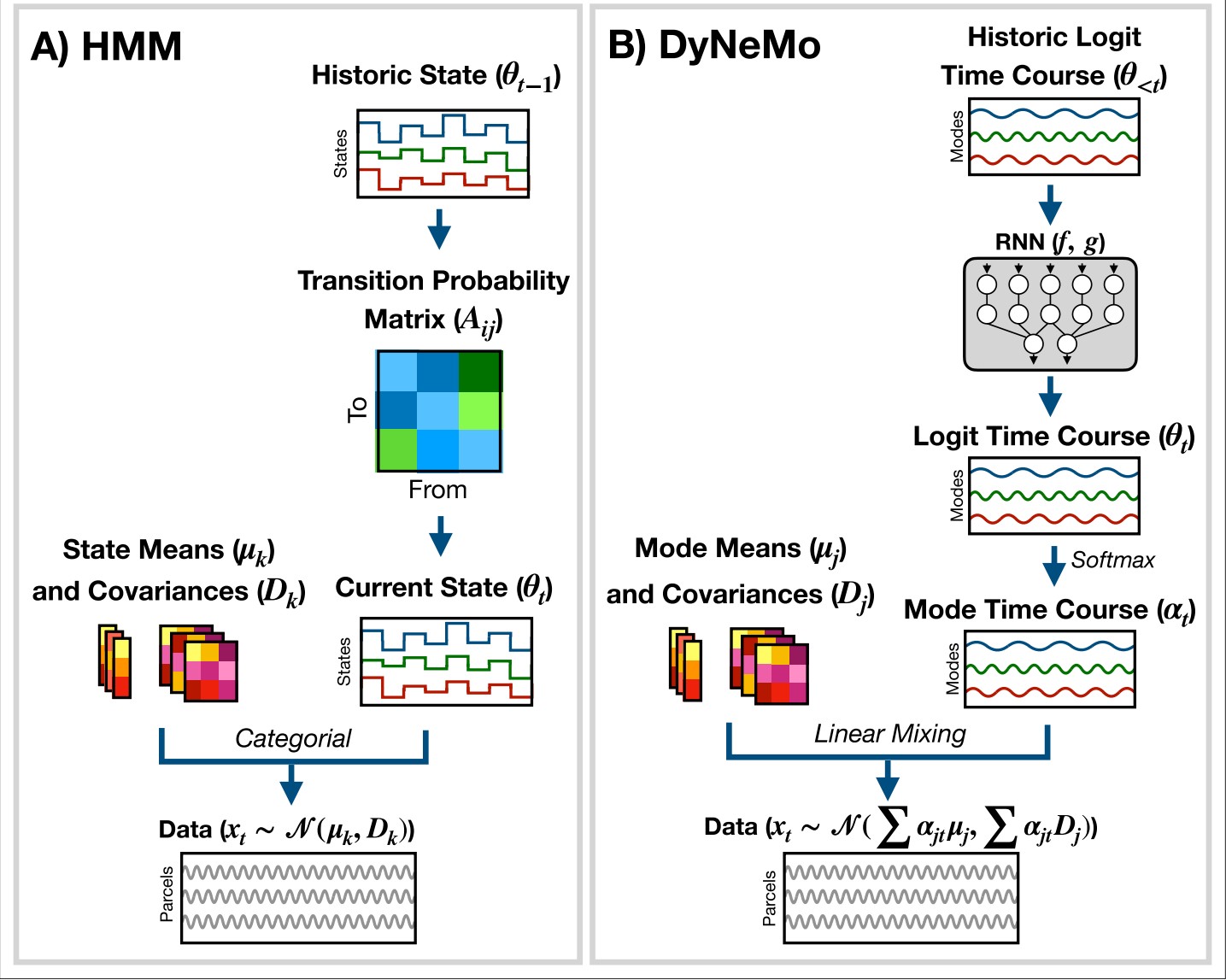

**Figure 7.** Generative models implemented in osl-dynamics. (**A**) Hidden Markov Model (HMM) (*Vidaurre et al., 2016*; *Vidaurre et al., 2018a*). Here, data is generated using a hidden state ($\theta_t$) and observation model, which in our case is a multivariate normal distribution parameterized by a state mean ($\mu_k$) and covariance ($D_k$). Only one state can be active at a given time point. Dynamics are modeled via state switching using a transition probability matrix ($A_{ij}$), which forecasts the probability of the current state based on the previous state. (**B**) Dynamic Network Modes (DyNeMo) (*Gohil et al., 2022*). Here, the data is generated using a linear combination of *modes* ($\mu_j$ and $D_j$) and dynamics are modelled using a recurrent neural network (RNN: $f$ and $g$), which forecasts the probability of a particular mixing ratio ($\alpha_t$) based on a long history of previous values via the underlying logits ($\theta_t$).

where $\theta_t \in \{1, ..., K\}$ is the latent state at time $t$, $K$ is the number of states, and $x_t$ is the generated data. $p(x_t|\theta_t)$ is the *observation model*. Here, we use

$$p(x_t|\theta_t = k) = \mathcal{N}(\mu_k, D_k), \qquad (2)$$

where $\mu_k \in \{\mu_1, ..., \mu_K\}$ is a state mean and $D_k \in \{D_1, ..., D_K\}$ is a state covariance. Dynamics in the time series are generated through state switching, which is characterized by the transition probability $p(\theta_t|\theta_{t-1})$. Each pairwise state transition forms the *transition probability matrix*, (*Rezek and Roberts, 2005*)

$$A_{ij} = p(\theta_t = j|\theta_{t-1} = i). \qquad (3)$$

osl-dynamics uses variational Bayesian inference *Bishop and Nasrabadi, 2006* to learn the most likely state to have generated the observed data. This has the advantage of being able to account for uncertainty in the latent state. For more information regarding the implementation of the HMM in osl-dynamics see the documentation: available here. The HMM has been successfully used to study dynamics in neuroimaging data in a variety of settings (*Seedat et al., 2020*; *Khawaldeh et al., 2022*; *Vidaurre et al., 2018b*; *Quinn et al., 2018*; *Higgins et al., 2021*; *Sitnikova et al., 2018*; *Van Schependom et al., 2019*; *Vidaurre et al., 2016*; *Vidaurre et al., 2018a*; *Vidaurre et al., 2017*).

DyNeMo is a recently proposed model that overcomes two key limitations of the HMM: the mutually exclusive states and limited memory (*Gohil et al., 2022*). The generative model for DyNeMo (shown in *Figure 7B*) is

$$p(\boldsymbol{x}_{1:T}, \boldsymbol{\theta}_{1:T}) = p(\boldsymbol{x}_1|\boldsymbol{\theta}_1)p(\boldsymbol{\theta}_1)\prod_{t=2}^{T} p(\boldsymbol{x}_t|\boldsymbol{\theta}_t)p(\boldsymbol{\theta}_t|\boldsymbol{\theta}_{1:t-1}), \tag{4}$$

where $\boldsymbol{\theta}_t$ is a latent vector at time $t$ (referred to as a *logit*) and $\boldsymbol{x}_t$ is the generated data. The observation model we use is

$$p(\boldsymbol{x}_t|\boldsymbol{\theta}_t) = \mathcal{N}(\boldsymbol{m}_t, \boldsymbol{C}_t),$$

$$\boldsymbol{m}_t = \sum_{j=1}^{J} \alpha_{jt}\boldsymbol{\mu}_j,$$

$$\boldsymbol{C}_t = \sum_{j=1}^{J} \alpha_{jt}\boldsymbol{D}_j, \tag{5}$$

where $\boldsymbol{\mu}_j \in \{\boldsymbol{\mu}_1, ..., \boldsymbol{\mu}_J\}$ is a mode mean, $\boldsymbol{D}_j \in \{\boldsymbol{D}_1, ..., \boldsymbol{D}_J\}$ is a mode covariance, $J$ is the number of modes and

$$\alpha_{jt} = \{\text{softmax}(\boldsymbol{\theta}_t)\}_j \tag{6}$$

is the mixing coefficient for mode $j$. Dynamics in the latent vector are generated through $p(\boldsymbol{\theta}_t|\boldsymbol{\theta}_{1:t-1})$, which is a distribution parameterized using a recurrent neural network (*Géron, 2022*). Specifically,

$$p(\boldsymbol{\theta}_t|\boldsymbol{\theta}_{1:t-1}) = \mathcal{N}(\boldsymbol{m}_{\theta_t}, \boldsymbol{\sigma}_{\theta_t}^2),$$

$$\boldsymbol{m}_{\theta_t} = f(\boldsymbol{\theta}_{1:t-1})$$

$$\boldsymbol{\sigma}_{\theta_t}^2 = g(\boldsymbol{\theta}_{1:t-1}), \tag{7}$$

where $f$ and $g$ are calculated using a recurrent neural network. osl-dynamics uses *amortized* variational Bayesian inference *Kingma and Welling, 2013* to learn the most likely latent vector to have generated the observed data. This is a highly efficient inference scheme that is scalable to large datasets. For more information regarding the implementation of DyNeMo in osl-dynamics see the documentation: available here.

Once trained, both models reveal a dynamic latent description of the training data. For the HMM, the latent description is a hidden state time course (Also known as the Viterbi path), which is the most likely state inferred at each time point in the training data. For DyNeMo, it is a mode time course, which is the mixing coefficient time series for each mode inferred from the training data. We will discuss in Sections 5.5.1 and 5.5.2 how these latent descriptions can be used to summarize dynamics in the training data.

## 5.2 Datasets
We make use of two publicly available datasets:

- CTF rest MEG dataset. This contains resting-state (eyes open) MEG data collected using a 275-channel CTF scanner. This dataset contains 5 min recordings from 65 healthy participants. It was collected at Nottingham University, UK as part of the MEGUK partnership (*Meg scientific research community, 2023*).
- Elekta task MEG dataset. This contains MEG data recorded during a visual perception task (*Wakeman and Henson, 2015*). Six runs from 19 healthy participants were recorded using an

Elekta Neuromag Vectorview 306 scanner. This dataset was collected at Cambridge University, UK.

## 5.3 Preprocessing and source reconstruction

The steps involved in estimating source data from an MEG recording are shown in *Figure 8*. This part of the pipeline can be performed with the OHBA Software Library (OSL) (*Mats, 2023*; *Quinn et al., 2022b*), which is a separate Python package for M/EEG analysis. The exact steps applied to the raw data for each dataset were:

1. MaxFilter (only applied to the Elekta dataset).
2. Bandpass filter 0.5–125 Hz.
3. Notch filter 50 Hz and 100 Hz.
4. Downsample to 250 Hz.
5. Automated bad segment removal and bad channel detection (See osl.preprocessing.osl_wrappers.detect_badsegments and detect_badchannels in *Mats, 2023*).
6. Automated ICA cleaning using the correlation of the EOG/ECG channel to select artifact components (See osl.preprocessing.mne_wrappers.run_mne_ica_autoreject in *Mats, 2023*).
7. Coregistration (using polhemus headshape points/fiducials and a structural MRI).
8. Bandpass filter 1–45 Hz.
9. Linearly Constrained Minimum Variance (LCMV) beamformer.
10. Parcellate to regions of interest. In this work, we used 38 parcels (We used a parcellation based on anatomy: fmri_d100_parcellation_with_PCC_reduced_2 mm_ss5mm_ds8mm.nii.gz *Mats, 2023*).
11. Symmetric orthogonalization (to correct source leakage *Colclough et al., 2015*).
12. Dipole sign flipping (to align the sign of parcel time courses across subjects/runs) (See osl.source_recon.sign_flipping in *Mats, 2023*. Note, this step can be skipped if you are training on amplitude envelope data).
13. Downsample to 100 Hz (only included in the burst detection pipeline).

These preprocessing steps have been found to work well for a wide variety of datasets when studying dynamics. The scripts used for preprocessing and source reconstruction can be found here: https://github.com/OHBA-analysis/osl-dynamics/tree/main/examples/toolbox_paper.

## 5.4 Data preparation

We usually prepare the source data before training a model. The data preparation can be different depending on what aspect of the data we are interested in studying.

Amplitude envelope (AE). If we are interested in studying dynamics in the amplitude of oscillations, we can train a model on AE data. Here, we typically bandpass filter a frequency range of interest and calculate an AE using the absolute value of a Hilbert transform. *Figure 9B* shows what happens when we calculate the AE of oscillatory data. We can see the AE data tracks changes in the amplitude of oscillations.

Time-delay embedding (TDE). Studying the amplitude dynamics of oscillations does not reveal any insights into how different regions interact via phase synchronization. For this, we need to prepare the data using TDE (*Strogatz, 2018*). This augments the time series with extra channels containing time-lagged versions of the original channels. *Figure 9CI* shows an example of this. To perform TDE, we need to specify the number of lagged channels to add (number of embeddings) and the lag to shift each additional channel by. In osl-dynamics, we always shift by one-time point, so we only need to specify the number of lags. By adding extra channels, we embed the autocorrelation function (ACF) of the original data (as well as the cross-correlation function) into the covariance matrix of the TDE data. This is illustrated in *Figure 9CII*. We plot the ACF taken from the TDE covariance matrix and the PSD (calculated using a Fourier transform) in *Figure 9CIII*. By using TDE data we make the covariance matrix sensitive to the frequency of oscillations in the original data. The covariance matrix is also sensitive to cross channel phase synchronization via the off-diagonal elements. Training on TDE data allows us to study dynamics in oscillatory amplitude and phase synchronization between channels. When we prepare TDE data, we are normally only interested in looking for dynamics in the auto/cross correlation function via the covariance matrix, so we fix the mean to zero in the generative model.

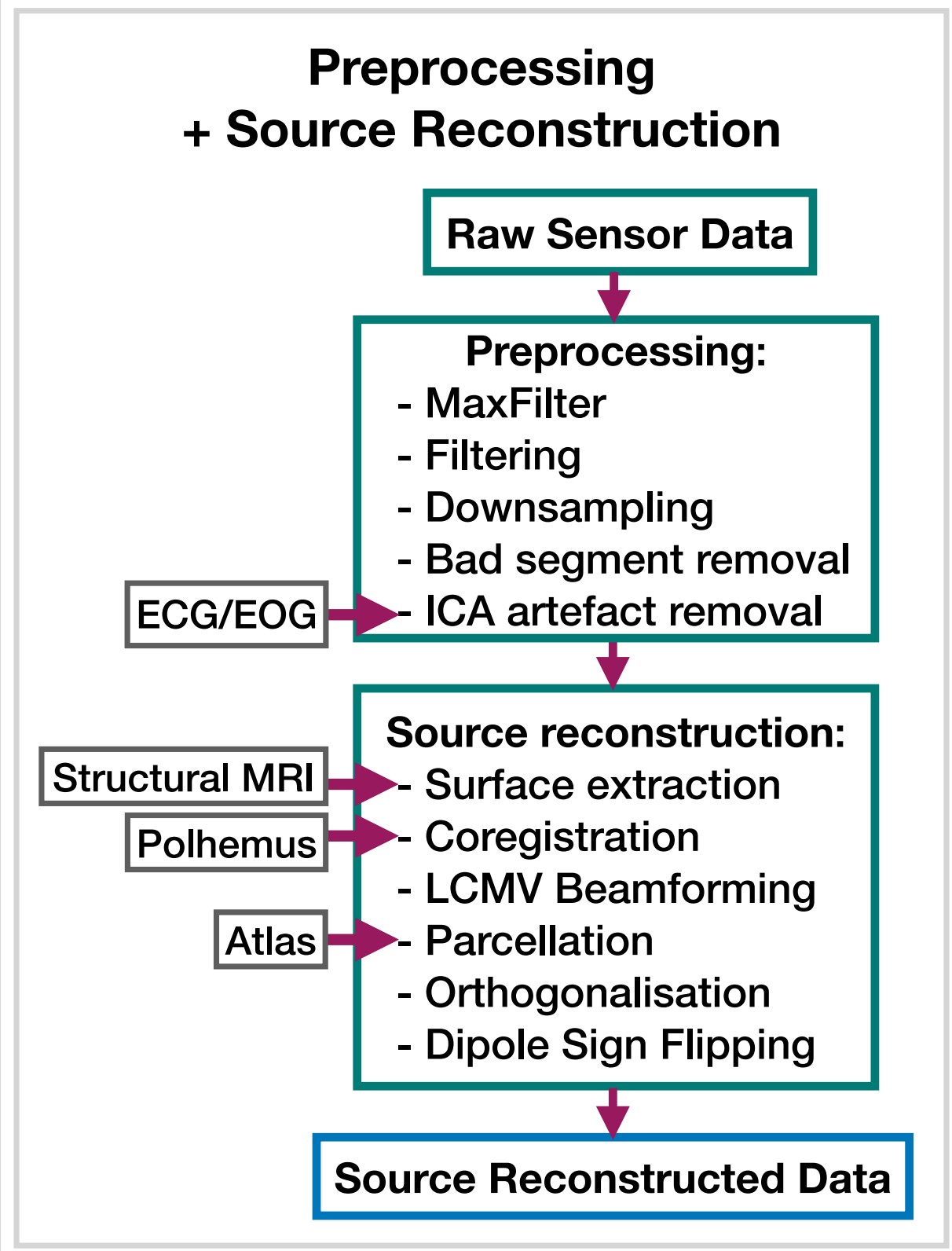

**Figure 8.** Preprocessing and source reconstruction. First, the sensor-level recordings are cleaned using standard signal processing techniques. This includes filtering, downsampling, and artifact removal. Following this, the sensor-level recordings are used to estimate source activity using a beamformer. Finally, we parcellate the data and perform corrections (orthogonalization and dipole sign flipping). Acronyms: electrocardiogram (ECG), electrooculogram (EOG), independent component analysis (ICA), linearly constrained minimum variance (LCMV). These steps can be performed with the OHBA Software Library: https://github.com/OHBA-analysis/osl, (copy archived at *OHBA-analysis, 2024*).

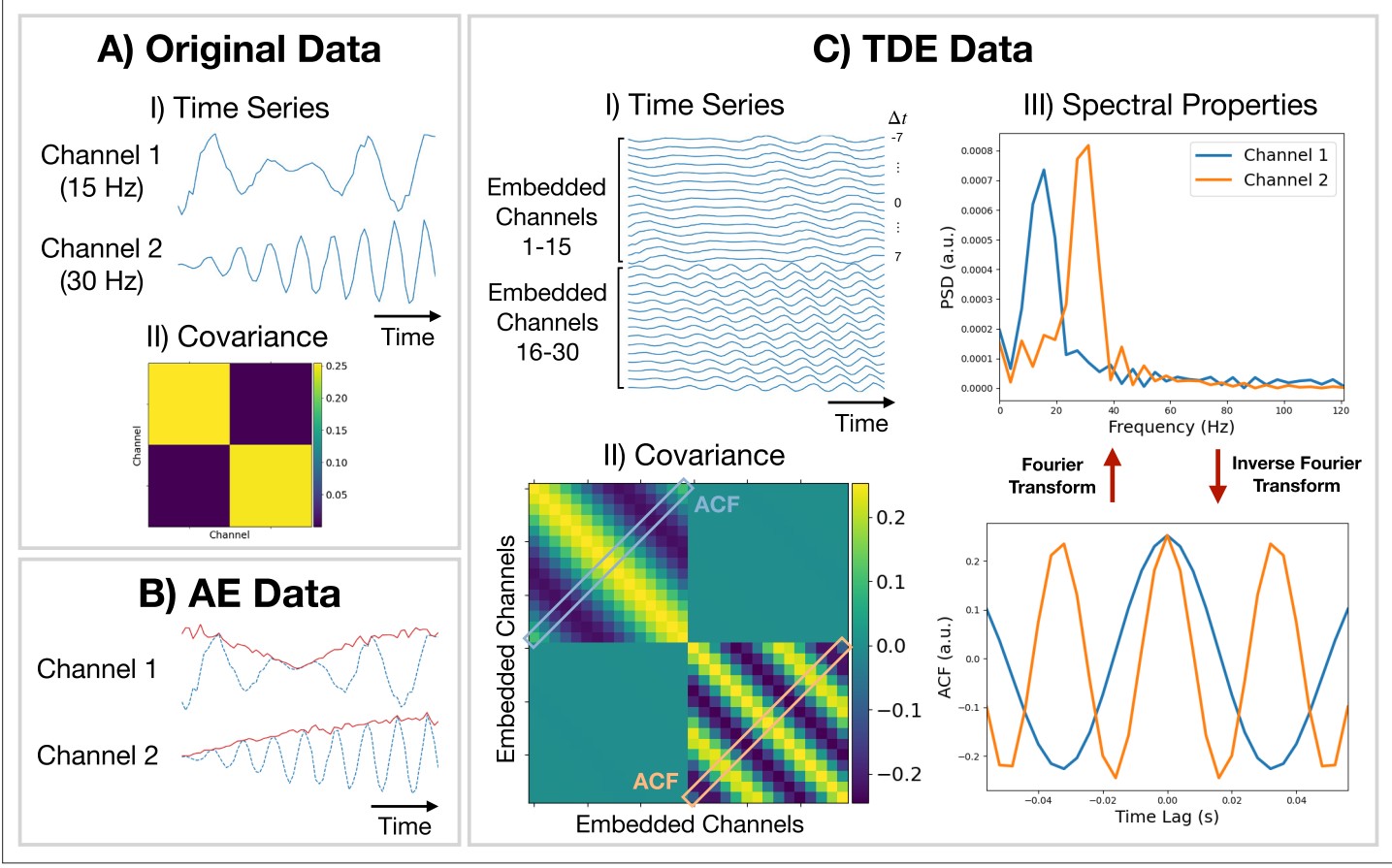

**Figure 9.** Methods for preparing training data. (**A.I**) Original (simulated) time series data. Only a short segment (0.2 s) is shown. Channel 1 (2) is a modulated sine wave at 15 Hz (30 Hz) with $\mathcal{N}(0, 0.1)$ noise added. (**A.II**) Covariance of the original data. (**B**) Amplitude envelope (AE) data (solid red line) and original data (dashed blue line). (**C.I**) Time-delay embedded (TDE) time series. An embedding window of ±7 lags was used. (**C.II**) Covariance of TDE data. (**C.III**) Spectral properties of the original data were estimated using the covariance matrix of TDE data. Acronyms: Autocorrelation function (ACF), Power spectral density (PSD).

For further details and example code for preparing data in osl-dynamics see the tutorial: available here.

## 5.5 First-level and group-level analysis

Starting from the source reconstructed data, we study a dataset with a two-stage process:

1. First-level analysis. Here, our objective is to estimate subject-specific quantities. In the static (time-averaged) analysis, we calculate these quantities directly from the source data. However, if we are doing a dynamic analysis, we first train a generative model, such as the HMM or DyNeMo. Note, the HMM/DyNeMo are typically trained as group-level models using the concatenated data from all subjects. This means the models are unaware that the data originates from different subjects and allows the model to pool information across subjects, which can lead to more robust estimates of dynamic quantities. We use the latent description provided by the model with the source data to estimate the quantities of interest - this approach is known as *dual estimation* (This step is analogous to dual regression in independent component analysis) (*Vidaurre et al., 2021*).
2. Group-level analysis. Quantities estimated for individual subjects, such as network metrics or summary statistics for dynamics, are used to model a group. For example, this could be predicting behavioral traits or characteristics of individual subjects, comparing two groups, or calculating the group average of network response to a task. Typically, statistical significance testing is done at the group level to verify that any observed differences or inferred relationships are not simply due to chance.

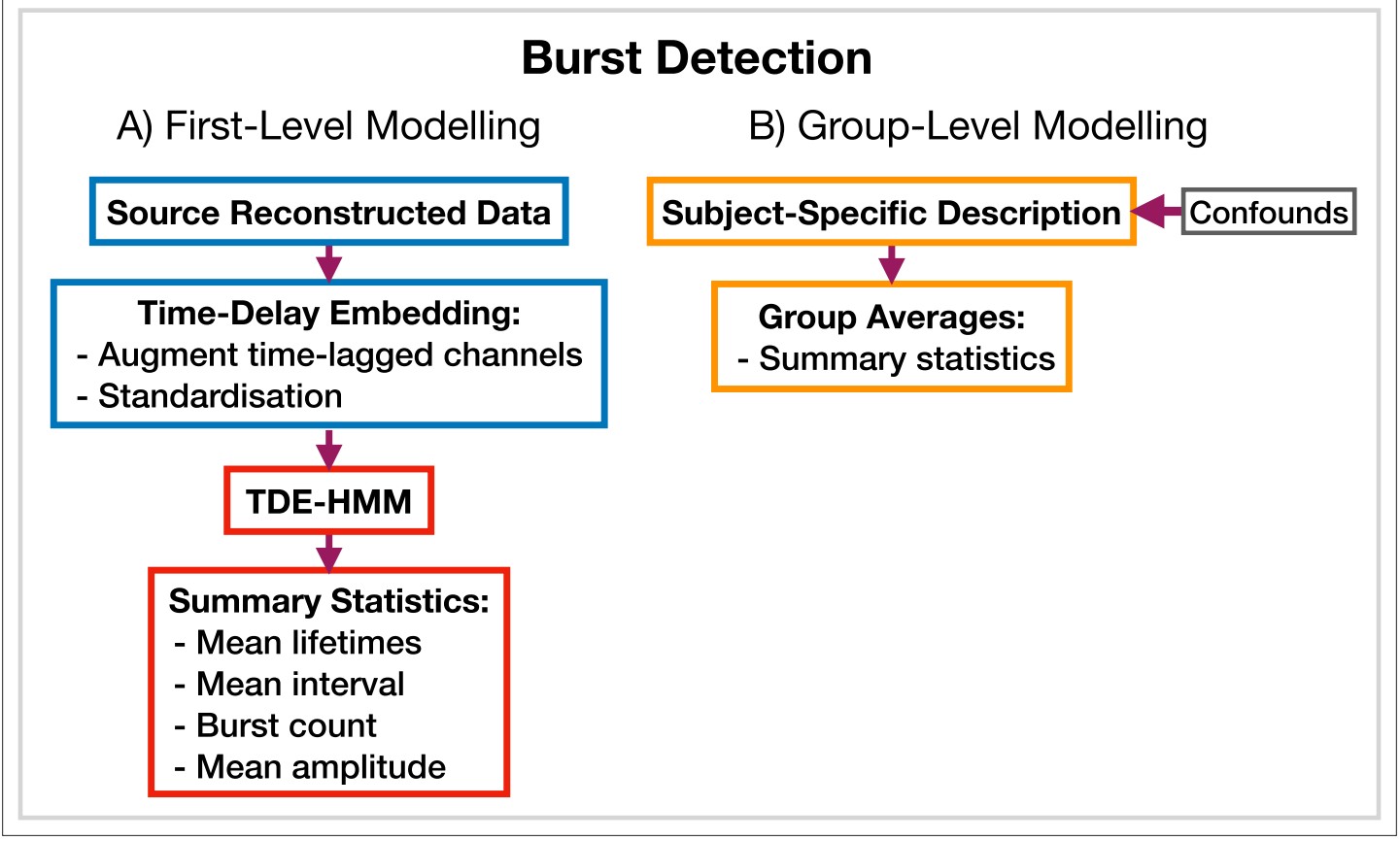

**Figure 10.** TDE-HMM burst detection pipeline. This is run on a single region's parcel time course. Separate Hidden Markov Models (HMMs) are trained for each region. (**A**) Source reconstructed data is prepared by performing time-delay embedding and standardization (z-transform). Following this an HMM is trained on the data and statistics that summarize the bursts are calculated from the inferred state time course. (**B**) Subject-specific metrics summarizing the bursts at a particular region are used in group-level analysis.

We will present the results of applying five pipelines to source reconstructed data calculated from the datasets mentioned in Section 5.2: a burst detection pipeline based on the HMM (discussed in Section 5.5.1); three dynamic network analysis pipelines based on the HMM and DyNeMo (discussed in Section 5.5.2) and a static network analysis pipeline (discussed in Section 5.5.3). Both the HMM and DyNeMo have been validated on simulated data for each application. See *Quinn et al., 2019* for a demonstration of the HMM's ability to identify oscillatory bursts and *Vidaurre et al., 2016*; *Gohil et al., 2022* for a demonstration of the HMM/DyNeMo's ability to identify dynamic networks.

### 5.5.1 Burst detection

We use an approach based on the HMM to detect bursts of oscillatory activity. In this approach, we prepare the source data using TDE. A typical TDE-HMM burst detection pipeline is shown in *Figure 10*. When the HMM state time courses are inferred from the training data, each 'visit' to a particular state corresponds to a burst, or transient spectral event, with spectral properties specific to the state (e.g. an increase in $\beta$-band power). This approach assumes that we are looking for bursting in a single channel (brain region) at a time; separate HMMs can be used to detect bursting in each channel. We use the state time course to calculate summary statistics that characterize the dynamics of bursts. Typical summary statistics are:

- Mean lifetime (The lifetime is also known as *dwell time*). This is the average duration a state is active.
- Mean interval. This is the average duration between successive state activations.
- Burst count. This is the number of times a state activates in a second on average.

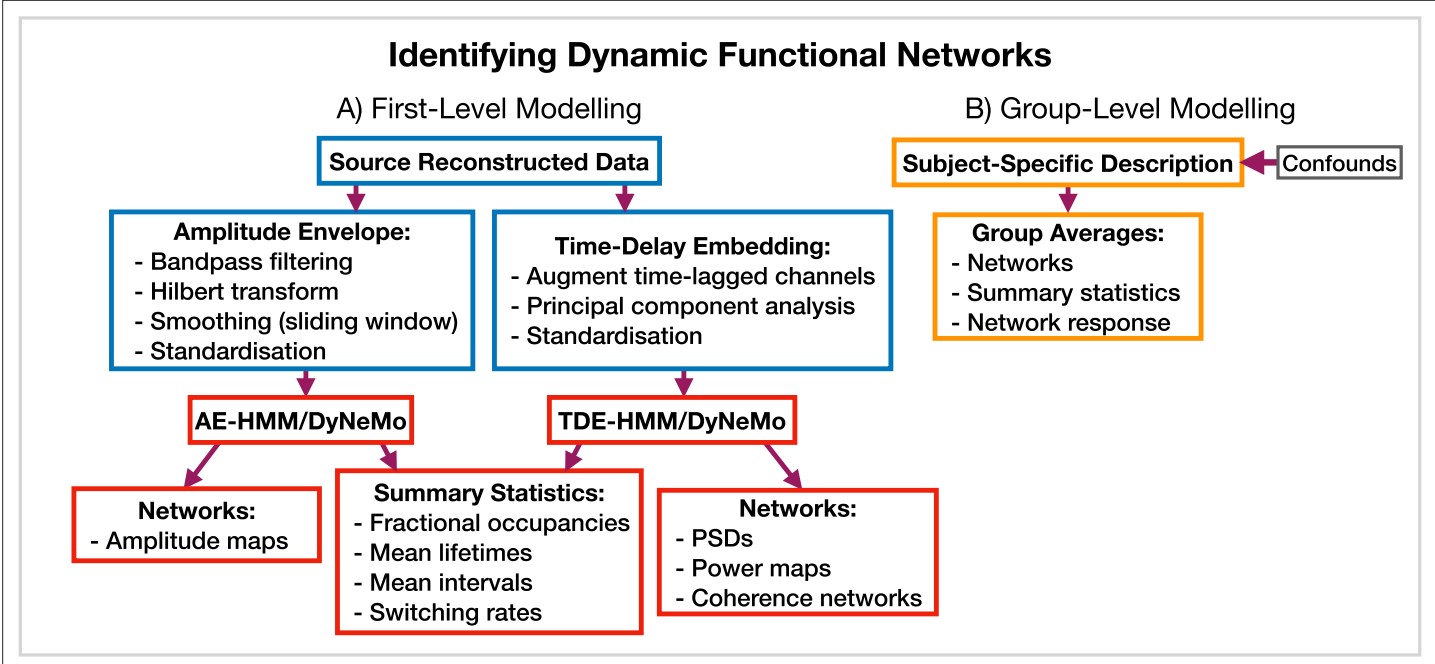

**Figure 11.** Dynamic functional network analysis pipeline. (**A**) First-level modeling. This includes data preparation (shown in the blue boxes), model training, and post-hoc analysis (shown in the red boxes). The first-level modelling is used to derive subject-specific quantities. (**B**) Group-level modelling. This involves using the subject-specific description from the first-level modeling to model a group.

- Mean amplitude. This is the average value of the AE of the source data when each state is active.

We calculate each of these for a particular state and subject. The averages are taken over all state activations. Given when a state is active we can use the source data to calculate the PSD of each burst type. We use the multitaper approach described in *Vidaurre et al., 2016* to do this due to its ability to accurately estimate spectra. We present the results of applying a TDE-HMM burst detection pipeline to the CTF rest MEG dataset in Section 2.1.

## 5.5.2 Identifying dynamic functional networks

osl-dynamics provides more options for modeling dynamic functional networks. Note, in this case, we train on multivariate data containing the activity at multiple regions of interest, rather than a single region, which is what we did in the burst detection pipeline (Section 5.5.1). Indeed, one perspective on using osl-dynamics to model dynamic functional networks, is that it is identifying bursts that span across multiple brain regions. *Figure 11* shows the different combinations of data preparation and generative models that are available for a dynamic network analysis pipeline. We discuss each of these options and when they should be used below.

AE-HMM. If we are interested in identifying dynamics in amplitude, we can train on AE data. Once we have trained a model, we can estimate subject and state-specific networks (amplitude maps) using the training data and inferred state time course. Additionally, we can calculate summary statistics that characterize the dynamics from the inferred state time course. These summary statistics are:

- Fractional occupancy. This is the fraction of the total time that each state is active.
- Mean lifetime. This is the average duration that a state is active.
- Mean interval. This is the average duration between successive state visits.
- Switching rate. This is the number of activations per second of a particular state.

We calculate each of these for a particular state and subject. The averages are taken over all state activations. We present the results of an AE-HMM pipeline on the Elekta task MEG dataset in Section 2.2.

TDE-HMM. We can use TDE data to study dynamics in phase synchronization as well as dynamics in amplitude. In a dynamic network analysis pipeline we train on a multivariate time series (i.e. the time series for all regions of interest together). This means after TDE we have a very large number of channels (number of embeddings times the number of regions). Therefore, we often need to perform principal component analysis (PCA) for dimensionality reduction to ensure the data fits into computer memory.

In the TDE-HMM pipeline, we can calculate the same summary statistics as the AE-HMM pipeline. However, to estimate the functional networks we use the multitaper approach described in *Vidaurre et al., 2016*. Here, we use the source data and inferred state time course to estimate subject, region and state-specific PSDs and cross PSDs. When then use the PSDs to calculate power maps and cross PSDs to calculate coherence networks, see *Vidaurre et al., 2016* for further details. Note that we also use the spectral decomposition approach introduced in *Vidaurre et al., 2018b* to specify a frequency range for calculating power maps and coherence networks. This involves applying non-negative matrix factorization to the stacked subject and state-specific coherence spectra to identify common frequency bands of coherent activity. In this report, we fit two spectral components and only present the networks for the first band, which typically covers 1–25 Hz. We will see the results of applying a TDE-HMM pipeline for dynamic network analysis on both the CTF rest and Elekta task MEG dataset in Section 2.3.

TDE-DyNeMo. This this pipeline, we replace the HMM with DyNeMo and train on TDE data. Unlike the mutually exclusive state description provided by the HMM, DyNeMo infers mode time courses, which describe the mixing ratio of each mode at each time point (*Gohil et al., 2022*). This mixture description complicates the calculation of subject-specific quantities, such as networks and summary statistics. To calculate mode and region-specific PSDs, we use the approach based on the General Linear Model (GLM) proposed in *Quinn et al., 2022a* where we regress the mixing coefficients onto a (cross) spectrogram, see *Gohil et al., 2022* for further details. We then use the mode PSDs and cross PSDs to calculate power maps and coherence networks, respectively. We can summarize the dynamics of each mode time course with quantities such as the mean, standard deviation, and pairwise Pearson correlation. Alternatively, if we were interested in calculating the same summary statistics as the HMM (fractional occupancy, lifetime, interval, switching rate) we would first need to binarise the mode time courses. This can be done using a two-component Gaussian Mixture Model (GMM), which is discussed in *Gohil et al., 2022*. Note that an additional complication related to the mode time course is that it does not contain any information regarding the relative magnitude of each mode covariance. For example, a mode with a small value for the mixing ratio can still be a large contributor to the

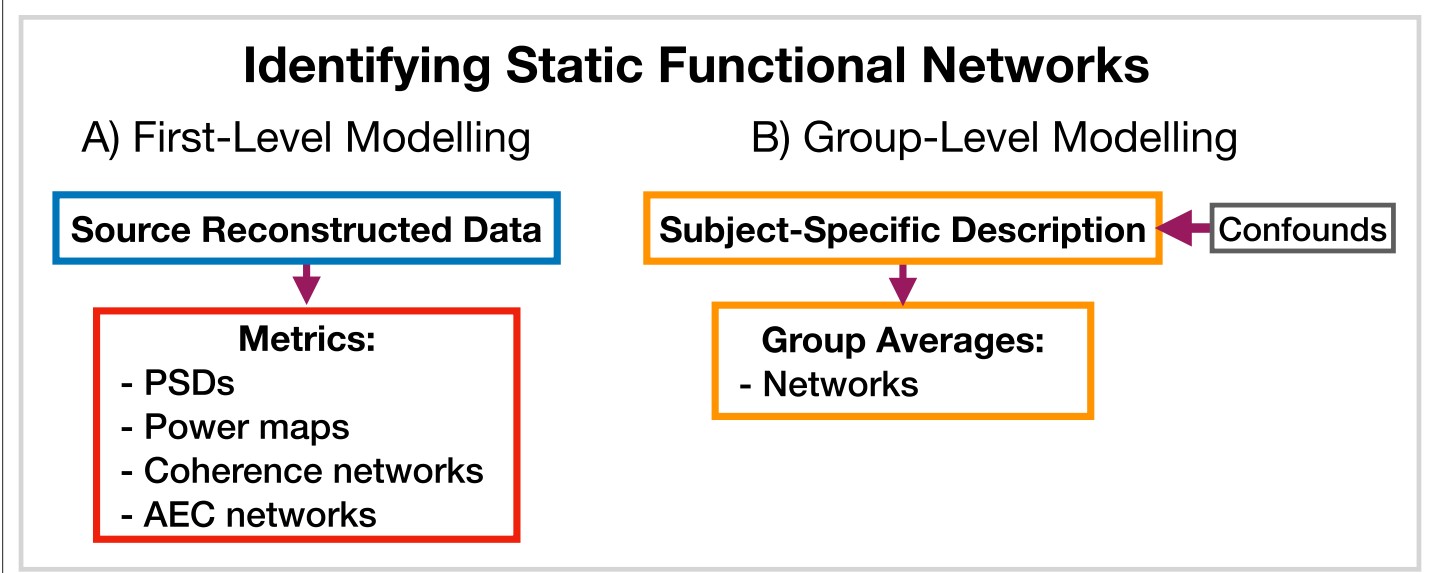

**Figure 12.** Static functional network analysis pipeline. (**A**) The source reconstructed data is used to calculate metrics that describe networks. (**B**) The subject-specific metrics are used to model a group. Acronyms: Amplitude envelope correlation (AEC), Power spectral density (PSD).

instantaneous covariance if the values in the mode covariance matrix are relatively large. We account for this by renormalizing the mode time course (We weigh each mode time course by the trace of its mode covariance and divide by the sum over modes at each time point to ensure the renormalized mode time course sums to one), this is discussed further in *Gohil et al., 2022*. We present the results of a TDE-DyNeMo pipeline on the CTF rest MEG dataset in Section 2.4.

AE-DyNeMo. The final option is to train DyNeMo on AE data. In this case, the amplitude maps are calculated using the GLM approach by regressing the mixing coefficients on a sliding window AE time course. Summary statistics for dynamics are calculated in the same way as the TDE-DyNeMo pipeline.

When we display the networks inferred by each of the pipelines above, we will threshold them to only show the strongest connections. In this work, we will specify the threshold using a data-driven approach where we fit a two-component GMM to the distribution of connections in each network. We interpret one of the components as the distribution for background connections and the other as the distribution for atypically strong connections, which is what we display in each plot.

### 5.5.3 Identifying static functional networks

A feature of osl-dynamics is that more conventional, static (time-averaged), network analyses can be carried out using the same methodology that we use in the dynamic methods. This allows for a much more straightforward comparison between static and dynamic analyses. To model static functional networks we simply need to specify the metrics we would like to use to summarize the networks and we calculate these directly from the source data. *Figure 12* shows a typical static network analysis pipeline. We present the result of a static network analysis pipeline on the CTF rest MEG dataset in Section 2.5. Note, for the static networks we select the top 5% of connections to display in each plot rather than the GMM approach we used to threshold the dynamic functional networks.

## 5.6 Run-to-run variability

The HMM and DyNeMo are trained by minimizing a cost function (in osl-dynamics, we use the *variational free energy Rezek and Roberts, 2005*; *Gohil et al., 2022*). As is typical, this approach suffers from a local optimum issue, where the model can converge to different explanations (latent descriptions) of the data during training. That is, different state/mode time courses can lead to similar values for the variational free energy. The final description can be sensitive to the stochasticity in updating model parameters and the initial parameter values.

A strategy for dealing with this that has worked well in the past is to train multiple models from scratch (each model is referred to as a *run*) and only the model with the lowest variational free energy is analyzed. We consider this model as the best description of the data. We ensure any conclusions based on this model are reproducible in the best model from another set of independent runs. In all of our figures here, we present the results from the best run from a set of 10. In the supplementary information (SI) we show these results are reproducible in an independent set of runs. Other strategies for dealing with run-to-run variability involve combining multiple runs, see *Alonso and Vidaurre, 2023* for a discussion of these techniques.

## 5.7 Reproducibility

We use variational Bayesian inference to learn model parameters (state/mode time courses and observation model means/covariances). This process involves updating the model parameters to minimize the *variational free energy* (*Bishop and Nasrabadi, 2006*).

Each time we train an HMM or DyNeMo , we start from random model parameters and use a stochastic procedure to update the parameters. This leads to some variability in the final model parameters we converge to and their corresponding variational free energy. We deem the model parameters with the lowest free energy as the ones that best describe the data and use these for subsequence analysis. We deem a set of results as reproducible if we are consistently able to infer the same model parameters from a dataset. Empirically, we find picking the best run from a set of 10 consistently finds the same set of model parameters. Note, for a given set of hyperparameters, only the relative difference between values for the variational free energy is important.

We show the variational free energy for three sets of 10 runs in *Figure 1—figure supplement 1A* – *Figure 5—figure supplement 1A*. In red, we highlight the variational free energy for the best run (i.e. the one with the lowest value). Within a set of runs, we see large variability in the variational

free energy which would reflect variability in the model parameters. Only looking at the best run across sets we see similar values for the variational free energy. In *Figure 1—figure supplement 1B – Figure 5—figure supplement 1B*, we see the summary statistics for the best run from each set are very similar indicating we infer the same dynamics in each best run. In *Figure 2—figure supplement 1C – Figure 5—figure supplement 1C*, we see we infer the same amplitude/power maps. In *Figure 2—figure supplement 1D – Figure 5—figure supplement 1D*, we see we observe the same results from the group-level analysis across all three sets.

## Acknowledgements

The Wellcome Centre for Integrative Neuroimaging is supported by core funding from the Wellcome Trust (203139/Z/16/Z) and the NIHR Oxford Health Biomedical Research Centre (NIHR203316). The views expressed are those of the authors and not necessarily those of the NIHR or the Department of Health and Social Care.

## Additional information

### Funding

| Funder | Grant reference number | Author |
|---|---|---|
| Wellcome Trust | 10.35802/215573 | Chetan Gohil<br>Mats WJ van Es<br>Mark W Woolrich |
| Engineering and Physical Sciences Research Council | EP/S02428X/1 | Rukuang Huang |
| Engineering and Physical Sciences Research Council | EP/L016044/1 | Evan Roberts |
| Wellcome Trust | 10.35802/106183 | Mats WJ van Es<br>Mark W Woolrich |
| Dementia Research UK | RG94383/RG89702 | Mats WJ van Es<br>Mark W Woolrich |
| Novo Nordisk Fonden | NNF19OC-0054895 | Diego Vidaurre |
| European Research Council | 10.3030/850404 | Diego Vidaurre |
| Independent Research Fund Denmark | 2034-00054B | Diego Vidaurre |
| NIHR Oxford Biomedical Research Centre | NIHR203316 | Mark W Woolrich |

The funders had no role in study design, data collection and interpretation, or the decision to submit the work for publication. For the purpose of Open Access, the authors have applied a CC BY public copyright license to any Author Accepted Manuscript version arising from this submission.

### Author contributions

Chetan Gohil, Conceptualization, Data curation, Software, Formal analysis, Validation, Investigation, Visualization, Methodology, Writing - original draft, Writing - review and editing; Rukuang Huang, Software, Validation, Methodology; Evan Roberts, Software; Mats WJ van Es, Software, Methodology; Andrew J Quinn, Methodology; Diego Vidaurre, Methodology, Writing - review and editing; Mark W Woolrich, Conceptualization, Data curation, Supervision, Methodology, Writing - review and editing

### Author ORCIDs

Chetan Gohil ⓘ https://orcid.org/0000-0002-0888-1207
Rukuang Huang ⓘ https://orcid.org/0000-0002-6545-7517
Mats WJ van Es ⓘ https://orcid.org/0000-0002-7133-509X
Diego Vidaurre ⓘ https://orcid.org/0000-0002-9650-2229

Reviewer #1 (Public Review): https://doi.org/10.7554/eLife.91949.3.sa1
Reviewer #2 (Public Review): https://doi.org/10.7554/eLife.91949.3.sa2
Author response https://doi.org/10.7554/eLife.91949.3.sa3

## Additional files

### Supplementary files
MDAR checklist

### Data availability
Raw sensor-level MEG recordings are publicly available: https://doi.org/10.18112/openneuro.ds000117.v1.0.5 (Elekta Task Dataset); https://meguk.ac.uk/database/ (CTF Rest Dataset, Nottingham). Source reconstructed data for both datasets is available here: https://osf.io/by2tc/. Scripts to train models and reproduce results are available here: https://github.com/OHBA-analysis/osl-dynamics/tree/main/examples/toolbox_paper (copy archived at *OHBA Analysis Group, 2024*).

The following dataset was generated:

| Author(s) | Year | Dataset title | Dataset URL | Database and Identifier |
|---|---|---|---|---|
| Gohil C | 2023 | OSL Dynamics Toolbox | https://osf.io/by2tc/ | Open Science Framework, by2tc |

The following previously published dataset was used:

| Author(s) | Year | Dataset title | Dataset URL | Database and Identifier |
|---|---|---|---|---|
| Wakeman DC, Henson RN | 2015 | Multisubject, multimodal face processing | https://doi.org/10.18112/openneuro.ds000117.v1.0.5 | OpenNeuro, 10.18112/openneuro.ds000117.v1.0.5 |

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
