## [Editor Report · eLife assessment]

The authors present a comprehensive set of tools to compactly characterize the time-frequency interactions across a network. The utility of the toolbox is **compelling** and demonstrated through a series of exemplar brain imaging datasets. This **fundamental** work adds to the repertoire of techniques that can be used to study high-dimensional data.

---

## [Referee Report · Reviewer #1 (Public Review)]

In their revised manuscript, the authors have addressed all the concerns raised earlier (written below for completeness).

Summary:

These types of analyses use many underlying assumptions about the data, which are not easy to verify. Hence, one way to test how the algorithm is performing in a task is to study its performance on synthetic data in which the properties of the variable of interest can be apriori fixed. For example, for burst detection, synthetic data can be generated by injected bursts of known durations, and checking if the algorithm can pick it up. Burst detection is difficult in the spectral domain since direct spectral estimators have high variance (see Subhash Chandran et al., 2018, J Neurophysiol). Therefore, detected burst lengths are typically much lower than injected burst lengths (see their Figure 3). This problem can be solved by doing burst estimation in the time domain itself, for example, using Matching Pursuit (MP). I think the approach presented in this paper would also work since this model is also trained on data in the time domain. Indeed, the synthetic data can be made more "challenging" by injecting multiple oscillatory bursts that are overlapping in time, for which a greedy approach like MP may fail. It would be very interesting to test whether this method can "keep up" as the data is made more challenging. While showing results from brain signals directly (e.g., Figure 7) is nice, it will be even more impactful if it is backed up with results obtained from synthetic data with known properties.

I was wondering about what kind of "synthetic data" could be used for the results shown in Figure 8-12 but could not come up with a good answer. Perhaps data in which different sensory systems are activated (visual versus auditory) or sensory versus movement epochs are compared to see if the activation maps change as expected? We see similarities between states across multiple runs (reproducibility analysis) and across tasks (e.g. Figure 8 vs 9) and even methods (Figure 8 vs 10), which is great. However, we should also expect emergence of new modes specific to sensory activation (say auditory cortex for an auditory task). This will allow us to independently check the performance of this method.

The authors should explain the reproducibility results (variational free energy and best run analysis) in the Results section itself, to better orient the reader on what to look for.

Page 15: the comparison across subjects is interesting, but it is not clear why sensory-motor areas show a difference and the mean lifetime of the visual network decreases. Can you please explain this better? The promised discussion in section 3.5 can be expanded as well.

---

## [Referee Report · Reviewer #2 (Public Review)]

Summary:

The authors have developed a comprehensive set of tools to describe dynamics within a single time-series or across multiple time-series. The motivation is to better understand interacting networks within the human brain. The time-series used here are from direct estimates of the brain's electrical activity; however the tools have been used with other metrics of brain function and would be applicable to many other fields.

Strengths:

The methods described are principled, based on generative probabilistic models.

This makes them compact descriptors of the complex time-frequency data.

Few initial assumptions are necessary in order to reveal this compact description.

The methods are well described and demonstrated within multiple peer reviewed articles.

This toolbox will be a great asset to the brain imaging community.

Weaknesses:

The only question I had (originally) was how to objectively/quantitatively compare different network models. This has now been addressed by the authors in the latest revision.

---

## [Author Response]

The following is the authors’ response to the original reviews.

We thank the reviewers for their feedback. Our response and a summary of the changes made to the manuscript are shown below. In addition to the changes made in response to the reviewer’s comments, we made the following changes to improve the manuscript:

We updated figures 8 and 9 using data with improved preprocessing and source reconstruction. We now also include graphical network plots. This helps in the cross method (figure 8 vs 9) and cross dataset (figure 9 vs 10) comparison.We added funding acknowledgments and a credit author statement.

**Reviewer #1 (Public Review):**
Summary:These types of analyses use many underlying assumptions about the data, which are not easy to verify. Hence, one way to test how the algorithm is performing in a task is to study its performance on synthetic data in which the properties of the variable of interest can be apriori fixed. For example, for burst detection, synthetic data can be generated by injected bursts of known durations, and checking if the algorithm is able to pick it up. Burst detection is difficult in the spectral domain since direct spectral estimators have high variance (see Subhash Chandran et al., 2018, J Neurophysiol). Therefore, detected burst lengths are typically much lower than injected burst lengths (see Figure 3). This problem can be solved by doing burst estimation in the time domain itself, for example, using Matching Pursuit (MP). I think the approach presented in this paper would also work since this model is also trained on data in the time domain. Indeed, the synthetic data can be made more "challenging" by injecting multiple oscillatory bursts that are overlapping in time, for which a greedy approach like MP may fail. It would be very interesting to test whether this method can "keep up" as the data is made more challenging. While showing results from brain signals directly (e.g., Figure 7) is nice, it will be even more impactful if it is backed up with results obtained from synthetic data with known properties.

We completely agree with the reviewer that testing the methods using synthetic data is an important part of validating such an approach. Each of the original papers that apply these methods to a particular application do this. The focus of this manuscript is to present a toolbox for applying these methods rather than to introduce/validate the methods themselves. For a detailed validation of the methods, the reader should see the citations. For example, the following paper introduces the HMM as a method for oscillatory burst detection:

A.J. Quinn, et al. “Unpacking transient event dynamics in electrophysiological power spectra”. Brain topography 32.6 (2019): 1020-1034. See figures 2 and 3 for an evaluation of the HMM’s performance in detecting single-channel bursts using synthetic data.

We have added text to paragraph 2 in section 2.5 to clarify this burst detection method has been validated using simulated data and added references.

I was wondering about what kind of "synthetic data" could be used for the results shown in Figure 8-12 but could not come up with a good answer. Perhaps data in which different sensory systems are activated (visual versus auditory) or sensory versus movement epochs are compared to see if the activation maps change as expected. We see similarities between states across multiple runs (reproducibility analysis) and across tasks (e.g. Figure 8 vs 9) and even methods (Figure 8 vs 10), which is great. However, we should also expect the emergence of new modes specific to sensory activation (say auditory cortex for an auditory task). This will allow us to independently check the performance of this method.

The following papers study the performance of the HMM and DyNeMo in detecting networks using synthetic data:

D. Vidaurre, et al. “Spectrally resolved fast transient brain states in electrophysiological data”. Neuroimage 126 (2016): 81-95. See figure 3 in this paper for an evaluation of the HMM’s performance in detecting oscillatory networks using simulation data.C. Gohil, et al. “Mixtures of large-scale dynamic functional brain network modes”. Neuroimage 263 (2022): 119595. See figures 4 and 5 for an evaluation of DyNeMo performance in detecting overlapping networks and long-range temporal structure in the data.

We have added text to paragraph 2 in section 2.5 to clarify these methods have been well tested on simulated data and added references.

The authors should explain the reproducibility results (variational free energy and best run analysis) in the Results section itself, to better orient the reader on what to look for.

Considering the second reviewer’s comments, we moved the reproducibility results to the supplementary information (SI). This means the reproducibility results are no longer part of the main figures/text. However, we have added some text to help the reader understand what aspects indicate the results are reproducible in section 2 of the SI.

Page 15: the comparison across subjects is interesting, but it is not clear why sensory-motor areas show a difference and the mean lifetime of the visual network decreases. Can you please explain this better? The promised discussion in section 3.5 can be expanded as well.

It is well known that the frequency and amplitude of neuronal oscillations changes with age. E.g. see the following review: Ishii, Ryouhei, et al. "Healthy and pathological brain aging: from the perspective of oscillations, functional connectivity, and signal complexity." Neuropsychobiology 75.4 (2018): 151-161. We observe older people have more beta activity and less alpha activity. These changes are seen in time-averaged calculations, i.e. the amplitude of oscillations are calculated using the entire time series for each subject.

The dynamic analysis presented in the paper provides further insight into how changes in the time-averaged quantities can occur through changes in the dynamics of frequency-specific networks. The sensorimotor network, which is a network with high beta activity, has a higher fractional occupancy. This indicates the change we observe in time-average beta power may be due to a longer amount of time spent in the sensorimotor network. The visual network, which is a network with high alpha activity, shows reduced lifetimes, which can explain the reduced time-averaged alpha activity seen with ageing.

We hope the improved text in the last paragraph of section 3.5 clarifies this. It should also be taken into account that the focus of this manuscript is the tools rather than an in-depth analysis of ageing. We use the age effect as an example of the potential analysis this toolbox enables.

**Reviewer #2 (Public Review):**
Summary:The authors have developed a comprehensive set of tools to describe dynamics within a single time-series or across multiple time-series. The motivation is to better understand interacting networks within the human brain. The time-series used here are from direct estimates of the brain's electrical activity; however, the tools have been used with other metrics of brain function and would be applicable to many other fields.Strengths:The methods described are principled, and based on generative probabilistic models.This makes them compact descriptors of the complex time-frequency data.Few initial assumptions are necessary in order to reveal this compact description.The methods are well described and demonstrated within multiple peer-reviewed articles.This toolbox will be a great asset to the brain imaging community.Weaknesses:The only question I had was how to objectively/quantitatively compare different network models. This is possibly easily addressed by the authors.

We thank the reviewer for his/her comments. We address the weaknesses in our response in the “Recommendations For The Authors” section.

**Reviewer #1 (Recommendations For The Authors):**
Figure 2 legend: Please add the acronym for LCMV also.

We have now done this.

Section 2.5.1 page 8: the pipeline is shown in Figure 4, not 3.

This has been fixed.

**Reviewer #2 (Recommendations For The Authors):**
This is a great paper outlining a resource that can be applied to many different fields. I have relatively minor comments apart from one.How does one quantitatively compare network descriptors (from DyNeMo and TDE-HMM for example)? At the moment the word 'cleaner' (P17) is used, but is there any non-subjective way? (eg Free energy/ cross validation etc). At the moment it is useful that one method gives a larger effect size (in a comparison between groups).. but could the authors say something about the use of these methods as more/less faithful descriptors of the data? Or in other words, do all methods generate datasets (from the latent space) that can be quantitatively compared with the original data?

In principle, the variational free energy could be used to compare models. However, because we use an approximate variational free energy (an exact measure is not attainable) for DyNeMo and an exact free energy for the HMM, it is possible that any differences we see in the variational free energy between the HMM and Dynemo are caused by the errors in its approximation. This makes it unreliable for comparing across models. That said, we can still use the variational free energy to compare within models. Indeed, we use the variational free energy for quantitative model comparisons when we select the best run to analyse from a set of 10.

One viable approach for comparing models is to assess their performance on downstream tasks. In this manuscript, examples of downstream tasks are the evoked network response and the young vs old group difference. We argue a better performance in the downstream task indicates a more useful model within that context. This performance is a quantitative measure. Note, there is no straightforward answer to which is the best model. It is likely different models will be useful for different downstream tasks.

In terms of which model provides a more faithful description of the data. The more flexible generative model for DyNeMo means it will generate more realistic data. However, this doesn’t necessarily mean it’s the best model (for a particular downstream task). Both the HMM and DyNeMo provide complementary descriptions that can be useful.

We have clarified the above in paragraph 5 of section 4.

Other comments:Footnote 6 - training on concatenated group data seems to be important. It could be more useful in the main manuscript where the limitations of this could be discussed.

By concatenating the data across subjects, we learn a group-level model. By doing this, we pool information across all subjects to estimate the networks. This can lead to more robust estimates. We have moved this footnote to the main text in paragraph 1 of section 2.5 and added further information.

In the TDE burst detection section- please expand on why/how a specific number of states was chosen.

As with the HMM dynamic network analysis, the number of states must be pre-specified. For burst detection, we are often interested in an on/off type segmentation, which can be achieved with a 2 state HMM. However, if there are multiple burst types, these will all be combined into a single ‘on’ state. Therefore, we might want to increase the number of states to model multiple burst types. 3 was chosen as a trade-off to stay close to the on/off description but allow the model to learn more than 1 burst type. We have added text discussing this in paragraph 4 of section 4.

Normally the value of free energy is just a function of the data - and only relative magnitude is important. I think figures (eg 7c) would be clearer if the offset could be removed.

We agree only the relative magnitude is important. We added text clarifying this in section 2 of the SI. We think it would still be worthwhile to include the offset so that future users can be sure they have correctly trained a model and calculated the free energy.

Related to the above- there are large differences in model evidence shown between sets. Yet all sets are the same data, and all parameter estimates are more or less the same. Could the authors account for this please (i.e. is there some other parameter that differentiates the best model in one set from the other sets, or is the free energy estimate a bit variable).

We would like to clarify only the model parameters for the best run are shown in the group-level analysis. This is the run with the lowest variational free energy, which is highlighted in red. We have now clarified this in the caption of each figure. The difference in free energy for the best runs (across sets) is relatively small compared to the variation across runs within a set. If we were to plot the model parameters for each of the 10 runs in a set, we would see more variability. We have now clarified this in section 2 of the SI.

Also note, the group analysis usually involves taking an average. Small differences in the variational free energy could reflect small differences in subject-specific model parameters, which are then averaged out, giving virtually identical group effects.

And related once again, if the data are always the same, I wonder if the free-energy plots and identical parameter estimates could be removed to free up space in figures?

The reproducibility results have now been moved to the supplementary information (SI).

When citing p-values please specify how they are corrected (and over what please eg over states, nodes, etc?). This would be useful didactically as I imagine most users will follow the format of the presentation in this paper.

We now include in the caption further details of how the permutation significance testing was done.

Not sure of the value of tiny power maps in 9C. Would consider making it larger or removing it?

The scale of these power maps is identical to part (A.I). We have moved the reproducibility analysis to the SI, enlarged the figure and added colour bars. We hope the values are now legible.

Figure 3. I think the embedding in the caption doesn't match the figure (+-5 vs +-7 lags). Would be useful to add in the units of covariance (cii).

The number of embeddings in the caption has been fixed. Regarding the units for the covariances, as this is simulated data there aren’t really any units. Note, there is already a colour bar to indicate the values of each element.

Minimize variational free energy - it may be confusing for some readers that other groups maximize the negative free energy. Maybe a footnote?

We thank the reviewer for their suggestion. We have added a footnote (1).

Final question- and related to the Magnetoencephalography (MEG) data presented. These data are projected into source space using a beamformer algorithm (with its own implicit assumptions and vulnerabilities). Would be interested in the authors' opinion on what is standing between this work and a complete generative model of the MEG data - i.e. starting with cortical electrical current sources with interactions modeled and a dynamic environmental noise model (i.e. packing all assumptions into one model)?

In principle, there is nothing preventing us from including the forward model in the generative model and training on sensor level MEG data. This would be a generative model starting from the dipoles inside the brain to the MEG sensors. This is under active research. If the reviewer is referring to a biophysical model for brain activity, the main barrier for this is the inference of model parameters. However, note that the new inference framework presented in the DyNeMo paper (Gohil, et al. 2022) actually makes this more feasible. Given the scope of this manuscript is to present a toolbox for studying dynamics with existing methods, we leave this topic as future work.